# SMILE: Extended Deep Submodular Function-Based Instruction and In-context Learning Demonstration Selection

Zihan Chen [1]  Chengshuai Shi [2]  Song Wang [3]  Jundong Li [1]  Cong Shen [1]

## Abstract

Prompt optimization is a key way to steer large language models when fine-tuning is impractical. However, instruction optimization (IO) and in-context learning (ICL) demonstration selection are often optimized separately and combined post hoc, implicitly assuming that a "best" instruction and a "best" demonstration set compose well. In practice, their interactions are strong, making such decoupled pipelines brittle. We propose SMILE, an efficient method that *jointly* selects instructions and demonstrations. Our key observation is that the ICL performance exhibits consistent diminishing returns across diverse instructions. Leveraging this structure, SMILE learns an instruction-conditioned surrogate aligned with LLM feedback and instantiates it as an Extended Deep Submodular Function that captures sample–sample coverage, sample–query relevance, and sample–instruction compatibility. SMILE then performs greedy, query-adaptive selection of the instruction–demonstration pair. Experiments on six datasets and multiple LLM backbones show that SMILE consistently outperforms IO-only, ICL-only, and existing joint baselines, supporting a context engineering view of prompting: jointly optimizing interacting components rather than tuning them in isolation. Code is available at: https://github.com/Chen-1031/SMILE

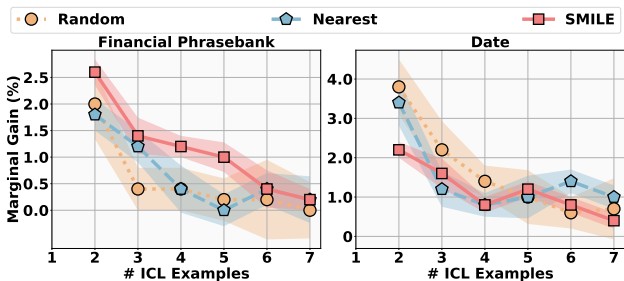

*Figure 1.* **Marginal gain vs. # ICL examples.** We plot the marginal gain $\Delta_k = \text{Perf}(k) - \text{Perf}(k-1)$ for $k \geq 2$, averaged over multiple fixed instructions $T$ (shaded: $\pm$ std; SMILE uses a fixed $T$ for each run), with Qwen3-4B-Instruct as the backbone LLM. All methods greedily add examples to form nested sets, so $\Delta_k$ measures the incremental benefit of the $k$-th example and highlights consistent diminishing returns across $T$.

## 1. Instruction

Large language models (LLMs) have achieved strong performance across a wide range of tasks, including mathematical problem solving (Liu et al., 2024; Bai et al., 2023), reasoning (Wei et al., 2022; Yao et al., 2023; Guo et al., 2025), and long-form summarization (Maharana et al., 2024; Bai et al., 2024). Yet their behavior remains highly sensitive to how the input is specified (Wan et al., 2024). This sensitivity has made prompt optimization a central mechanism for steering LLMs, particularly when fine-tuning is impractical (e.g., due to cost) or impossible (e.g., black-box APIs). Two dominant prompt-optimization paradigms have emerged. In-context learning (ICL) augments the prompt with a small set of labeled examples (demonstrations) so the model can infer the task format and decision boundary from context (Chen et al., 2025a; Singh et al., 2025; Agarwal et al., 2024; Li et al., 2025). Instruction optimization (IO) searches over alternative instructions (Xiang et al., 2025; Agrawal et al., 2025), typically by generating candidate instructions using LLM calls (or a labor-intensive process) and selecting the best-performing ones (Shi et al., 2024).

Despite their shared goal, ICL and IO techniques are often developed and evaluated in isolation. Wan et al. (2024) suggests that combining IO with ICL can yield gains beyond either component alone. However, existing approaches typically optimize instructions and demonstrations separately and then combine them post hoc (Wan et al., 2024), implicitly assuming that "good instructions" and "good demonstrations" compose. Yet instructions and demonstrations are tightly coupled, so optimizing them independently can

[1]University of Virginia, Charlottesville, VA, USA [2]Princeton University, Princeton, NJ, USA [3]University of Central Florida, Orlando, FL, USA. Correspondence to: Cong Shen <cong@virginia.edu>.

*Proceedings of the 43rd International Conference on Machine Learning*, Seoul, South Korea. PMLR 306, 2026. Copyright 2026 by the author(s).

be misleading: a demonstration set that is optimal for one instruction may underperform for another, and an instruction that works well in isolation may fail when paired with different demonstrations.

This motivates *joint selection*, but it is difficult because prompt components interact in complex ways. In this paper, we identify a structural regularity that we leverage to make principled progress on joint prompt selection. Empirically, ICL typically exhibits diminishing returns: adding demonstrations tends to help initially, but the marginal benefit decreases as the set grows (Agarwal et al., 2024; Bertsch et al., 2025; Baek et al., 2025). In Figure 1, we show that this diminishing-returns pattern is consistent across instructions: for a fixed model and demonstration pool, additional demonstrations yield decreasing marginal improvements under a wide range of instructions. While different instructions can change which demonstrations are most helpful, the shape of improvement, strong early gains followed by diminishing returns, remains stable. This regularity motivates modeling ICL utility with an (approximately) submodular surrogate conditioned on the instruction.

But translating this structure into a practical selection objective is nontrivial. Existing "submodular" ICL methods typically rely on hand-crafted objectives, such as embedding similarity to the query (Liu et al., 2022) or coverage measured by facility location in representation space (Kumari et al., 2024). While these objectives are mathematically submodular, they are *not* derived from the diminishing-return behavior of ICL performance itself, and therefore may be misaligned with what the LLM actually rewards. To turn the observed diminishing returns into a usable joint optimization principle, one must address two intertwined goals: (i) the surrogate should be *aligned* with the real ICL performance (rather than a convenient but potentially mismatched proxy), and (ii) it must capture the *interaction* between instructions and demonstrations, since instruction–demonstration compatibility can change which examples help and makes "optimize separately, then combine" unreliable.

Motivated by this gap, we propose SMILE, which learns a surrogate objective that is trained to match the ICL diminishing-return property while explicitly modeling instruction–demonstration interactions. SMILE instantiates an Extended Deep Submodular Function (EDSF) (Hosseini et al., 2024) to score demonstration sets conditioned on an instruction. The compositional structure of EDSF allows us to incorporate interpretable components that capture complementary interactions: (i) sample–sample interactions to encourage informative coverage within the selected set, (ii) sample–query interactions to capture relevance between demonstrations and the test input, and (iii) sample–instruction interactions to measure compatibility between demonstrations and the instruction, reducing con-

tradictory formats or patterns. We train the surrogate using feedback from the target LLM so that it aligns with observed ICL utility and exhibits approximate diminishing returns; despite not being perfectly submodular everywhere, we provide theoretical justification that greedy selection remains effective even when exact submodularity does not strictly hold. At inference time, SMILE pairs each candidate instruction with its selected demonstration set and chooses the best instruction–demonstration combination for the query. Our main contributions are summarized as follows:

- ***Joint prompt optimization.*** We propose SMILE, an EDSF-based surrogate that models the diminishing-returns structure of ICL while explicitly capturing instruction–demonstration interactions, enabling coherent joint selection.

- ***Theory for efficient selection.*** We provide theoretical justification showing that greedy selection remains effective for our learned surrogate, even when exact submodularity does not strictly hold.

- ***Strong empirical results.*** Extensive experiments across task types and models demonstrate consistent gains over competitive baselines, including settings with black-box proprietary LLMs, supporting the robustness and generality of the approach.

## 2. Methodology

### 2.1. Preliminaries

Given a finite size-$n$ set of objects $\mathcal{S}$, where each $s \in \mathcal{S}$ is a distinct element. A valuation set function $f : 2^{\mathcal{S}} \to \mathbb{R}$ is said to be *submodular* if for all $\mathcal{A} \subseteq \mathcal{B}$ and $s \notin \mathcal{B}$ the following inequality holds: $f(\mathcal{A} \cup \{s\}) - f(\mathcal{A}) \geq f(\mathcal{B} \cup \{s\}) - f(\mathcal{B})$ (Krause & Golovin, 2014). This means that the incremental value of adding another sample $s$ to a subset decreases when the context in which $s$ is considered grows from $\mathcal{A}$ to $\mathcal{B}$. Function $f$ is called a *modular* function if $f(\mathcal{A}) = \sum_{a \in \mathcal{A}} f(a), \ \forall \mathcal{A} \subseteq \mathcal{S}$. A function might also have the property of being normalized ($f(\emptyset) = 0$) and monotone non-decreasing ($f(\mathcal{A}) \leq f(\mathcal{B})$ whenever $\mathcal{A} \subseteq \mathcal{B}$). Choosing an appropriate submodular form is non-trivial, since the space of submodular functions is extremely large and cannot be searched or specified directly (Bilmes & Bai, 2017). This motivates restricting attention to a structured subfamily that is expressive yet learnable from data. In this work, we focus on a structured family termed extended deep submodular functions (Hosseini et al., 2024).

**Definition 2.1** (Deep Submodular Function[1]). Let $\mathcal{S}$ be a finite ground set. A *Deep Submodular Function* (DSF)

---

[1]The general definition of DSFs (Bilmes & Bai, 2017) includes an arbitrary modular term $m_{\pm}$. We omit these terms to ensure monotonicity, satisfying the prerequisites for the greedy approximation guarantee.

of depth $L$ is a set function $f : 2^{\mathcal{S}} \to \mathbb{R}$ defined by the following recursive structure:

1. **Base Layer** ($L = 1$): The nodes $\varphi_i^{(1)}$ are defined as a sum of concave composed with modular (SCM) functions:

$$\varphi_i^{(1)}(\mathcal{A}) = \sum_{j=1}^{n} \phi_{i,j}(m_j(\mathcal{A})), \quad \mathcal{A} \subseteq \mathcal{S}$$

where $m_j$ are non-negative modular functions and $\phi_{i,j}$ are normalized, non-decreasing, and concave.

2. **Recursive Step** ($L = \ell > 1$): Given the outputs of the previous layer $\Phi^{(\ell-1)}(\mathcal{A}) = [\varphi_1^{(\ell-1)}, \ldots, \varphi_{k_{\ell-1}}^{(\ell-1)}]$, the nodes of layer $\ell$ are defined as:

$$\varphi_v^{(\ell)}(\mathcal{A}) = \phi_v \left( \sum_{u=1}^{k_{\ell-1}} w_{uv} \varphi_u^{(\ell-1)}(\mathcal{A}) \right)$$

where $w_{uv} \in \mathbb{R}_+$ are non-negative weights and $\phi_v$ are non-decreasing concave functions.

By induction, every node in a depth-$L$ DSF is a valid monotone submodular function (Bilmes & Bai, 2017).

**Definition 2.2** (Extended Deep Submodular Function (EDSF)). A set function $h : 2^{\mathcal{S}} \to \mathbb{R}$ is an *Extended Deep Submodular Function* if there exist $r \geq 1$ deep submodular functions $f_1, \ldots, f_r$ such that for all $\mathcal{A} \subseteq \mathcal{S}$,

$$h(\mathcal{A}) = \min\{ f_1(\mathcal{A}), f_2(\mathcal{A}), \ldots, f_r(\mathcal{A}) \}.$$

We restate a key theoretical result from Hosseini et al. (2024), which establishes the expressive coverage of the EDSF family over (monotone) submodular functions. This motivates EDSF as a surrogate for capturing the diminishing-returns behavior observed in ICL.

**Theorem 2.3** (Expressiveness of EDSFs). *The family of monotone (submodular) set functions coincides exactly with the family of Extended Deep Submodular Functions.*

### 2.2. Problem Setting

Given a set of labeled examples $\mathcal{D} = \{e_i\}_{i=1}^{N}$, where each example $e_i = (x_i, y_i)$ consists of an input–output pair, a set of task instructions $\mathcal{T} = \{T_j\}_{j=1}^{n}$, and a test set $\mathcal{D}_{\text{test}}$. For each test example $(x, y) \in \mathcal{D}_{test}$ and an in-context learning budget $K$, our goal is to select an instruction $T^* \in \mathcal{T}$ with a set of demonstrations $\mathcal{S}^* \in \{\mathcal{S} | \mathcal{S} \subseteq \mathcal{D}, |\mathcal{S}| = K\}$, which serves as the input conditioning for a pretrained LLM $\mathcal{M}$ to make predictions on $x$ that maximizing the reward:

$$T^*, \mathcal{S}^* = \arg\max_{T, \mathcal{S}} \mathcal{R}(\mathcal{M}(T, \mathcal{S}, x), y). \quad (1)$$

where $\mathcal{R}(\cdot, \cdot)$ denotes some task-specific scoring function, for example, accuracy for classification tasks.

### 2.3. Overview

Directly optimizing the instruction $T$ and demonstration set $\mathcal{S}$ for each query $q$ via LLM feedback at inference time is impractical, as it would require many expensive LLM calls to evaluate candidate prompts (Shi et al., 2024). To avoid this cost, SMILE learns a *performance surrogate* that preserves the diminishing-returns structure of ICL while remaining easy to optimize (Figure 2). Specifically, leveraging the expressivity of EDSF, we define $g(q, T, \mathcal{S}) = s_{\text{QI}}(q, T) + f(\mathcal{S}; q, T)$, where $s_{\text{QI}}(q, T)$ scores query–instruction compatibility and $f$ is a parameterized EDSF surrogate that predicts the utility of $\mathcal{S}$ conditioned on $(q, T)$. We train $f$ using LLM feedback on the training set while preserving a set-function form amenable to efficient optimization (e.g., greedy selection). With this surrogate, we rewrite Eq. 1 as the following optimization: given a query $q$, select $(T, \mathcal{S})$ to maximize

$$(T^*, \mathcal{S}^*) = \arg\max_{T \in \mathcal{T}} \left[ s_{\text{QI}}(q, T) + \max_{\mathcal{S}} f(\mathcal{S}; q, T) \right]. \quad (2)$$

Our surrogate $f(\mathcal{S}; q, T)$ captures three types of prompt interactions via expert submodular functions: sample–sample ($f^{\text{SS}}$), sample–query ($f^{\text{SQ}}$), and sample–instruction ($f^{\text{SI}}$) (Section 2.4). Following Definition 2.2, we aggregate them as $f(\mathcal{S}; q, T) = \min\{f^{\text{SS}}, f^{\text{SQ}}, f^{\text{SI}}\}$. To optimize Eq. 2, we first select $\mathcal{S}_T = \arg\max_{|\mathcal{S}|=K} f(\mathcal{S}; q, T)$ for each $T \in \mathcal{T}$, and then choose $T^* = \arg\max_{T \in \mathcal{T}} [s_{\text{QI}}(q, T) + f(\mathcal{S}_T; q, T)]$, setting $\mathcal{S}^* = \mathcal{S}_{T^*}$.

### 2.4. EDSF Surrogate for Demonstration Selection

We build an EDSF surrogate to score a demonstration set $\mathcal{S}$ given instruction $T$ and query $q$. The surrogate combines three experts capturing complementary interactions, each using inexpensive per-example signals and a concave-over-modular form to model diminishing returns.

**Sample–Sample Expert** ($f^{\text{SS}}$). A core principle for ICL is that the demonstration set should provide broad coverage over the task's semantic modes, rather than repeatedly selecting near-duplicate examples (Su et al., 2022; Chen et al., 2024a). We therefore instantiate the sample–sample expert using modular *bucket-coverage* channels and concave saturations to induce diminishing returns. Specifically, we form $k$ semantic buckets by applying $k$-means to demonstration embeddings $\mathbf{h}_i = \text{Emb}(x_i)$, yielding centroids $\{\mathbf{c}_t\}_{t=1}^{k}$. For each bucket $t$, we define a non-negative soft assignment weight for each example $e_i$:

$$a_{i,t} \triangleq \frac{\exp(\cos(\mathbf{h}_i, \mathbf{c}_t))}{\sum_{j : e_j \in \mathcal{D}} \exp(\cos(\mathbf{h}_j, \mathbf{c}_t))} \in (0, 1),$$

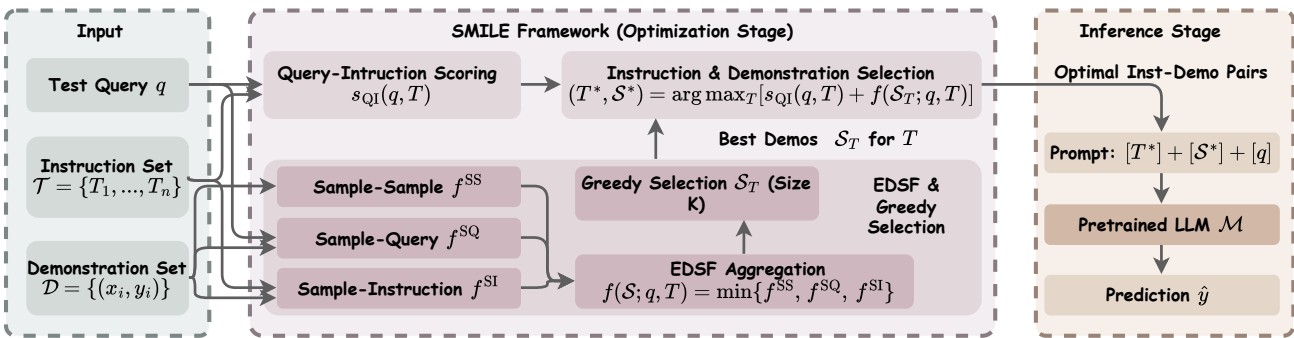

*Figure 2.* **SMILE framework.** For a query $q$, SMILE scores candidate instructions $T \in \mathcal{T}$ with $s_{\mathrm{QI}}(q, T)$ and greedily selects $K$ demonstrations $\mathcal{S}_T \subset \mathcal{D}$ by maximizing an instruction-conditioned EDSF surrogate. It then outputs the best pair $(T^*, \mathcal{S}^*)$ and prompts the LLM with $(T^*, \mathcal{S}^*, q)$ to generate the final answer.

and induce the modular coverage channel

$$m_t(\mathcal{S}) = \sum_{e_i \in \mathcal{S}} a_{i,t}, \qquad t \in [k].$$

Intuitively, $m_t(\mathcal{S})$ measures how much bucket-$t$ affinity mass is captured by $\mathcal{S}$; maximizing concave saturations of $\{m_t\}$ encourages $\mathcal{S}$ to cover many buckets while avoiding redundant concentration on a single bucket.

**Sample–Query Expert ($f^{\mathrm{SQ}}$).** Demonstrations that are more relevant to the query often yield larger gains (Liu et al., 2022). Thus, our sample–query expert focuses on measuring how relevant a candidate demonstration is to the query $q$. We instantiate $f^{\mathrm{SQ}}$ with two query-conditioned modular channels, and then aggregate them with concave saturations to capture diminishing returns.

**(i) Semantic relevance.** Let $\mathbf{h}_q$ denote the query embedding and $\mathbf{h}_i$ the embedding of example $e_i$. We define semantic relevance with cosine similarity score:

$$m_{\mathrm{sim}}^q(\mathcal{S}) = \sum_{e_i \in \mathcal{S}} \frac{1 + \cos(\mathbf{h}_i, \mathbf{h}_q)}{2}. \qquad (3)$$

**(ii) Lexical relevance.** To provide a model-agnostic lexical signal, we compute the ROUGE-L F1 score between the query $q$ and each candidate example $e$:

$$m_{\mathrm{rougeL}}^q(\mathcal{S}) = \sum_{e_i \in \mathcal{S}} \mathrm{ROUGE\text{-}L\text{-}F1}(x_i, q). \qquad (4)$$

These channels provide complementary relevance signals (semantic and lexical) and can be aggregated with concave saturations to discourage redundancy.

**Sample–Instruction Expert ($f^{\mathrm{SI}}$).** While query relevance and set coverage are widely studied, the interaction between demonstrations and the instruction is crucial for *joint* selection: a demonstration that is useful under one

instruction may be misleading under another (e.g., mismatched format or contradictory reasoning pattern). We therefore prefer demonstrations that are *effective under the instruction $T$*. We instantiate $f^{\mathrm{SI}}$ using two complementary modular channels: an example-level *instruction effect* and a prototype-based *applicability prior*. For each candidate example $e_i = (x_i, y_i)$, we define:

**(i) Information gain.** We quantify how much the instruction $T$ increases the model's preference for the ground-truth output under a zero-shot setting. Specifically, we compute teacher-forced log-likelihood normalized by output length:

$$\log p(y_i \mid T, x_i) \triangleq \frac{1}{|y_i|} \sum_{t=1}^{|y_i|} \log p(y_{i,t} \mid T, x_i, y_{i,<t}),$$

and analogously for a default instruction $T_0$. We then define

$$\begin{aligned} m_{\mathrm{IG}}^T(\mathcal{S}) &= \sum_{e_i \in \mathcal{S}} \mathrm{IG}(e_i; T) \\ &= \sum_{e_i \in \mathcal{S}} \log p(y_i \mid T, x_i) - \log p(y_i \mid T_0, x_i). \end{aligned}$$

This provides an example-specific estimate of the *instruction effect*: $\mathrm{IG}(e_i; T)$ is high when $T$ substantially improves the normalized likelihood of the correct output compared to $T_0$.

**(ii) Prototype similarity.** While informative, $\mathrm{IG}(\cdot; T)$ can be noisy (e.g., due to stochasticity in model scoring). To obtain a smoother notion of which *types* of examples benefit from $T$, we form an instruction prototype as a softmax-weighted average of embeddings:

$$\mathbf{p}_T \triangleq \sum_{i: e_i \in \mathcal{D}} \frac{\exp(\mathrm{IG}(e_i; T))}{\sum_{k: e_k \in \mathcal{D}} \exp(\mathrm{IG}(e_k; T))} \mathbf{h}_i.$$

We then define

$$m_{\mathrm{proto}}^T(\mathcal{S}) = \sum_{e_i \in \mathcal{S}} \frac{1 + \cos(\mathbf{h}_i, \mathbf{p}_T)}{2}.$$

$m_{\mathrm{proto}}^T$ captures a region-level applicability prior: examples whose embeddings lie near the prototype are preferred,

improving stability and generalization beyond noisy per-example scores.

**DSF Aggregation within Experts and EDSF Instantiation.** Each expert is instantiated using the same DSF template. Given non-negative modular channels $\{m_u(\mathcal{S})\}_u$ with $m_u(\mathcal{S}) = \sum_{e \in \mathcal{S}} s_u(e)$ and $s_u(e) \geq 0$, we aggregate them via concave saturations:

$$f(\mathcal{S}) \triangleq \sum_u w_u \, \phi_u\big(m_u(\mathcal{S})\big),$$

where $w_u \geq 0$ and $\phi_u(\cdot)$ is non-decreasing and concave (e.g., $\log(1+x)$ or $\min\{x, \delta\}$). This concave-over-modular construction yields a monotone submodular set function and implements diminishing returns (Bilmes & Bai, 2017). We apply this template to: (i) $\{m_t\}_{t=1}^k$ for $f^{\mathrm{SS}}$, (ii) $\{m_{\mathrm{sim}}^q, m_{\mathrm{rougeL}}^q\}$ for $f^{\mathrm{SQ}}$, and (iii) $\{m_{\mathrm{IG}}^T, m_{\mathrm{proto}}^T\}$ for $f^{\mathrm{SI}}$. Finally, we combine the three expert scores into the overall EDSF surrogate used for demonstration selection[2]:

$$f(\mathcal{S}; q, T) \triangleq \min\Big\{ f^{\mathrm{SS}}(\mathcal{S}), \; f^{\mathrm{SQ}}(\mathcal{S}; q), \; f^{\mathrm{SI}}(\mathcal{S}; T) \Big\}.$$

The resulting surrogate favors sets that are *jointly* strong across coverage, query relevance, and instruction compatibility, rather than over-optimizing any single aspect. Since each expert is a concave-over-modular DSF, it exhibits diminishing returns; moreover, the EDSF family is expressive enough to model monotone submodular utilities (Hosseini et al., 2024), matching the empirical diminishing-returns behavior of ICL. In Section 2.6, we show that greedy selection remains effective and efficient for this learned surrogate under mild approximate-submodularity conditions.

## 2.5. Calibration, Training Objective, and Inference

After defining our EDSF surrogate $f(\mathcal{S}; q, T)$, we describe how we *learn* its lightweight calibration parameters so that the surrogate aligns with the true ICL performance, and how we *use* the learned surrogate at inference time. Concretely, all modular channels (e.g., embedding similarity, information gain) are fixed; we only learn non-negative calibration weights $\mathbf{w} = \{w_u\}_u$ that scale these channels. We train a separate $\mathbf{w}$ for each downstream task using a small held-out development set and a fixed budget of LLM reward evaluations.

**Training Data Construction.** Let $\mathcal{D}_{\mathrm{dev}}$ be a held-out development set, $\mathcal{T}$ an instruction pool, and $\mathcal{D}$ a demonstration pool. We collect supervision under a budget of $B$ reward evaluations, where each evaluation corresponds to one triple $(q, T, \mathcal{S})$. At each sampling step, we draw:

---

[2]SMILE is *not* restricted to these specific choices: any per-example scoring signal like length-based cost can be plugged into our framework.

$q = (x, y) \sim \mathrm{Unif}(\mathcal{D}_{\mathrm{dev}})$, $T \sim \mathrm{Unif}(\mathcal{T})$, and an ICL budget $K \sim \mathrm{Unif}(\mathcal{K})$ with $\mathcal{K} = \{1, 2, 4, 8, \dots\}$. We then sample a seed set $\mathcal{S}_0 \subseteq \mathcal{D}$ with $|\mathcal{S}_0| = K$ and generate two local variants: a *swap* neighbor $\mathcal{S}_{\mathrm{swap}}$ obtained by replacing one element in $\mathcal{S}_0$ with a random element from $\mathcal{D} \setminus \mathcal{S}_0$ (so $|\mathcal{S}_{\mathrm{swap}}| = K$), and a *drop* neighbor $\mathcal{S}_{\mathrm{drop}}$ obtained by removing one element from $\mathcal{S}_0$ (so $|\mathcal{S}_{\mathrm{drop}}| = K - 1$). Thus, each sampling step yields three triples $\{(q, T, \mathcal{S}_0), (q, T, \mathcal{S}_{\mathrm{swap}}), (q, T, \mathcal{S}_{\mathrm{drop}})\}$. For each triple $(q, T, \mathcal{S})$, we compute a dense reward using normalized teacher-forced log-likelihood of the gold target $r(q, T, \mathcal{S}) = \log p(y \mid T, \{(x_i, y_i)\}_{e_i \in \mathcal{S}}, x)$. All evaluated tuples are stored in a buffer $\mathcal{B}$.

From $\mathcal{B}$, we derive pairwise preference constraints within the *same* query: *(i) within-instruction* constraints $(q, T, \mathcal{S}_a) \succ (q, T, \mathcal{S}_b)$ whenever $r(q, T, \mathcal{S}_a) > r(q, T, \mathcal{S}_b)$, which train set ranking conditioned on a fixed instruction; and *(ii) cross-instruction* constraints $(q, T, \mathcal{S}) \succ (q, T', \mathcal{S}')$ whenever $r(q, T, \mathcal{S}) > r(q, T', \mathcal{S}')$, which supports joint selection over $T$. Cross-instruction constraints require no extra LLM calls: they are formed by offline pairing of cached evaluations for the same query. Note that the set size is implicit in $\mathcal{S}$ and the training signal therefore also captures budget sensitivity. Applying the above construction yields a preference set $\mathcal{P}$ for training.

**Training Objective and Optimization.** Our EDSF surrogate uses a robust conjunction over experts, $f(\mathcal{S}; q, T) = \min\{f^{\mathrm{SS}}, f^{\mathrm{SQ}}, f^{\mathrm{SI}}\}$. However, the hard minimum can lead to winner-takes-all subgradients and unstable optimization. We therefore train with a fixed-temperature smooth relaxation (softmin):

$$\tilde{f}_\tau(\mathcal{S}; q, T) = -\tau \log \sum_{\mathrm{ex} \in \{\mathrm{SS}, \mathrm{SQ}, \mathrm{SI}\}} \exp\big(-f^{\mathrm{ex}}(\mathcal{S}; q, T)/\tau\big),$$

where $\tau > 0$ is held constant, ensuring gradient flow to all experts while preserving the robust behavior.

Let $\mathbf{w}$ denote the learnable calibration weight. We fit $\mathbf{w}$ using a logistic pairwise ranking loss over preferences in $\mathcal{P}$:

$$\mathcal{L}_{\mathrm{rank}}(\mathbf{w}) = \frac{1}{|\mathcal{P}|} \sum_{((q, T, \mathcal{S}^+), (q, T', \mathcal{S}^-))} \log\Big(1 + \exp\big(-\Delta_{\mathbf{w}}\big)\Big),$$

$$(5)$$

$$\Delta_{\mathbf{w}} \triangleq \tilde{f}_\tau(\mathcal{S}^+; q, T) - \tilde{f}_\tau(\mathcal{S}^-; q, T'), \qquad (6)$$

and add $\ell_2$ regularization to prevent any single component from dominating:

$$\mathcal{L}(\mathbf{w}) = \mathcal{L}_{\mathrm{rank}}(\mathbf{w}) + \beta \|\mathbf{w}\|_2^2. \qquad (7)$$

To enforce $w_u \geq 0$, we parameterize $w_u = \mathrm{softplus}(\psi_u)$ and optimize $\psi$ on pairs sampled from $\mathcal{P}$.

**Joint Selection at Inference.** At test time, we revert to the original hard-min composition $f(\cdot; q, T)$, since it is the surrogate we ultimately optimize for selection. Given a query $q$ and an ICL budget $K$, we select $\mathcal{S}_T = \arg\max_{|\mathcal{S}|=K} f(\mathcal{S}; q, T)$ by greedy selection for each instruction $T \in \mathcal{T}$. We then choose the final instruction–demonstration pair by scoring each instruction with the selected set:

$$(T^*, \mathcal{S}^*) = \arg\max_{T \in \mathcal{T}} \Big( s_{\mathrm{QI}}(q, T) + f_{\mathbf{w}}(\mathcal{S}_T; q, T) \Big),$$

where $s_{\mathrm{QI}}(q, T)$ is the $m_{\mathrm{proto}}(q)$ score (defined in Section 2.4) used to capture query-dependent instruction preference without additional LLM calls.

## 2.6. Efficiency Analysis

We analyze the efficiency of SMILE from two perspectives: (i) the effectiveness of greedy selection for our learned surrogate and (ii) the computational overhead of SMILE.

**Greedy selection for the learned surrogate.** For monotone submodular objectives, greedy achieves a $(1 - 1/e)$ approximation under a cardinality constraint (Krause & Golovin, 2014). In practice, the exact submodularity assumptions may not hold everywhere for the true ICL utility (and hence for a learned surrogate). We therefore analyze greedy selection under a mild approximate-submodularity/per-step progress condition, which we empirically observe to hold for our learned surrogate (Figure 1). We begin with a standard property of monotone submodular functions.

**Lemma 2.4** (Good-step condition under submodularity). *Let $f : 2^{\mathcal{D}} \to \mathbb{R}_+$ be a monotone submodular function and let $f^\star := \max_{\mathcal{A} \subseteq \mathcal{D}, |\mathcal{A}| \leq k} f(\mathcal{A})$. For any $\mathcal{A} \subseteq \mathcal{D}$ with $|\mathcal{A}| < k$, there exists an element $x \in \mathcal{D} \setminus \mathcal{A}$ such that*

$$f(\mathcal{A} \cup \{x\}) - f(\mathcal{A}) \geq \tfrac{1}{k}\big(f^\star - f(\mathcal{A})\big). \quad (8)$$

The next theorem shows that greedy remains near-optimal as the above progress condition holds *most of the time*, even when the objective is not strictly submodular everywhere.

**Theorem 2.5** (High-probability guarantee for greedy under per-step distortion). *Let $f : 2^{\mathcal{D}} \to \mathbb{R}_+$ be a monotone set function and let $f^\star := \max_{\mathcal{A} \subseteq \mathcal{D}, |\mathcal{A}| \leq k} f(\mathcal{A})$. Consider greedy, which constructs $\mathcal{A}_0 = \varnothing, \mathcal{A}_1, \ldots, \mathcal{A}_k$ by adding one element per iteration. Suppose that at each iteration $t \in \{1, \ldots, k\}$, with probability at least $1 - \varepsilon$ (possibly conditional on past history), the good-step condition in equation 8 holds. Let $\delta \in (0, 1)$. Then, with probability at least $1 - \delta$, the greedy solution $\mathcal{A}_k$ satisfies*

$$\frac{f(\mathcal{A}_k)}{f^\star} \geq 1 - \exp\Big(-1 + \varepsilon + \sqrt{2\ln(1/\delta)/k}\Big). \quad (9)$$

The bound in Equation 9 recovers the classical $(1 - 1/e)$ behavior when $\varepsilon$ is small and $k$ is moderate/large. In Appendix E, we further discuss the small-$k$ regime and provide

additional interpretations. Together, these results justify using greedy maximization as an effective and computationally efficient way for demonstration selection with SMILE.

**Computation and Memory Cost.** We now discuss the overhead introduced by SMILE. Compared to standard similarity-based ICL baselines, most channels in our surrogate are query-independent and can be precomputed offline (the main query-dependent cost comes from the sample–query expert), so the online overhead is modest beyond computing embeddings. Concretely, we store one instruction prototype $\mathbf{p}_T \in \mathbb{R}^d$ for each $T \in \mathcal{T}$ and (optionally) cache example-level instruction effects used by the sample–instruction expert. This requires $\mathcal{O}(|\mathcal{T}|d)$ memory for prototypes plus $\mathcal{O}(|\mathcal{T}|N)$ scalars if we cache per-example instruction scores, where $N = |\mathcal{D}|$. Since $|\mathcal{T}|$ is typically small, this storage is mild in our settings. When $|\mathcal{T}|$ is large, we avoid full enumeration by prefiltering a query-specific subset $\mathcal{T}_q$ of size $K_T \ll |\mathcal{T}|$ using the cheap query–instruction score (e.g., $m_{\mathrm{proto}}^T(q)$). SMILE then performs demonstration selection only for $T \in \mathcal{T}_q$, reducing instruction evaluations from $\mathcal{O}(|\mathcal{T}|)$ to $\mathcal{O}(K_T)$ per query.

# 3. Experiments

## 3.1. Experimental Settings

**Baselines.** We compare SMILE against three baseline groups. **(i) Instruction optimization (IO).** We include a vanilla baseline using prompts in Appendix G, as well as two representative IO methods: APE (Zhou et al., 2022) and OPRO (Yang et al., 2024). **(ii) In-context learning (ICL).** Following Wan et al. (2024), we evaluate: *Random*, which uniformly samples demonstrations; *Diversity*, which clusters the demonstration pool with $k$-means and selects examples closest to the cluster centroids; and *Nearest*, which retrieves the most similar demonstrations to the test query by cosine similarity in a text-embedding space. **(iii) IO+ICL combinations.** We consider APE+R.S., which applies APE for IO and uses random search for demonstration selection; this combination is reported to be a strong baseline in Wan et al. (2024). We also include MIPRO (Opsahl-Ong et al., 2024), a representative joint optimization method. We provide the baseline details in Appendix C.

**Datasets and LLMs.** We evaluate on six datasets spanning five tasks: **(i) Summarization** on XSum (Narayan et al., 2018); **(ii) Algorithmic reasoning** on Date and Salient following Lee et al. (2024); **(iii) Sentiment analysis** on Financial PhraseBank (FP) (Malo et al., 2014); **(iv) Question answering** on GPQA (Rein et al., 2023); and **(v) Math** on GSM8K (Cobbe et al., 2021). We report ROUGE-L F1 on XSum, exact match on GSM8K, and accuracy on the remaining datasets. Detailed descriptions of the datasets used are provided in Appendix D. Our main experiments

*Table 1.* **Main results on six datasets with two LLMs.** We compare SMILE with eight baselines, grouped as instruction optimization (IO; red), ICL demonstration selection (ICL; blue), and IO+ICL joint optimization (green). Higher is better (exact match for GSM8K, ROUGE-L for XSum, and accuracy for the remaining tasks). Bold and underline denote the best and second-best results, respectively.

| Method | GSM8K | GPQA | Financial Phrasebank | XSum | Date | Salient | Avg. |
|---|---|---|---|---|---|---|---|
| *Qwen3-4B-Instruct* | | | | | | | |
| **Vanilla** | 82.4 | 20.7 | 92.4 | 16.9 | 36.8 | 59.6 | 51.5 |
| **APE** (Zhou et al., 2022) | 88.0 | 32.8 | 87.2 | 15.8 | 46.2 | 54.2 | 54.0 |
| **OPRO** (Yang et al., 2024) | 89.2 | 32.4 | 89.6 | 17.9 | 58.0 | 54.0 | 56.8 |
| **Random** (Wan et al., 2024) | 88.8 | 33.3 | 90.8 | 17.1 | 50.4 | 53.6 | 55.7 |
| **Diversity** (Wan et al., 2024) | 90.4 | 33.3 | 94.0 | 17.8 | 51.6 | 54.8 | 57.0 |
| **Nearest** (Wan et al., 2024) | 90.8 | 36.9 | 94.4 | 18.2 | 49.6 | 64.4 | 59.0 |
| **APE+R.S.** (Wan et al., 2024) | 87.2 | 37.4 | 93.2 | 18.5 | 56.4 | 66.4 | 59.8 |
| **MIPRO** (Opsahl-Ong et al., 2024) | 90.4 | 34.7 | 93.4 | 18.6 | 55.2 | 64.8 | 59.5 |
| **SMILE** | **91.2** | **38.9** | **96.0** | 18.5 | **63.2** | **70.0** | **63.0** |
| *Llama3.1-8B-Instruct* | | | | | | | |
| **Vanilla** | 82.4 | 22.3 | 89.2 | 19.1 | 57.6 | 44.0 | 52.4 |
| **APE** (Zhou et al., 2022) | 82.8 | 19.2 | 94.0 | 18.7 | 60.4 | 51.2 | 54.4 |
| **OPRO** (Yang et al., 2024) | 83.6 | 22.7 | 93.2 | 16.5 | 62.0 | 52.8 | 55.1 |
| **Random** (Wan et al., 2024) | 83.2 | 18.2 | 91.2 | 21.9 | 47.6 | 48.6 | 51.8 |
| **Diversity** (Wan et al., 2024) | 84.8 | 22.7 | 92.4 | 22.4 | 54.6 | 49.8 | 54.5 |
| **Nearest** (Wan et al., 2024) | 84.8 | 23.2 | 90.8 | 22.8 | 54.5 | 55.2 | 55.2 |
| **APE+R.S.** (Wan et al., 2024) | 83.6 | 28.2 | **94.8** | 20.2 | 59.2 | 58.4 | 57.4 |
| **MIPRO** (Opsahl-Ong et al., 2024) | 82.4 | 23.7 | 90.8 | 21.1 | 60.2 | 62.0 | 56.7 |
| **SMILE** | **85.2** | **30.3** | 92.8 | **22.8** | **66.0** | **67.2** | **60.7** |

use Qwen3-4B-Instruct (Yang et al., 2025) and Llama3.1-8B-Instruct (Dubey et al., 2024) as backbone LLMs, and we further test generalization to black-box API models (GPT-5.2 (OpenAI) and Gemini2.5-Flash (Comanici et al., 2025)).

**Implementation.** In the main experiments, we uniformly sample 400 examples as the demonstration pool $\mathcal{D}$ and 100 examples as a held-out development set $\mathcal{D}_{\text{eval}}$. For evaluation, if the original test split contains more than 250 instances, we uniformly sample 250; otherwise, we use the full test split. We set the ICL budget to $K = 10$, and the default prompts and $T_0$ for computing information gain are provided in Appendix G. We use Qwen3-Embedding-0.6B (Zhang et al., 2025) as the embedding model $\text{Emb}(\cdot)$. For the EDSF surrogate, we set the concave function to $\phi(x) = \log(1 + x)$, and report an ablation with $\phi(x) = \min\{x, \delta\}$ in Appendix A. We set the number of channels in $f^{SS}$ to 10 to match the *Diversity* ICL baseline. We set the instruction set size to $|\mathcal{T}| = 10$, where $\mathcal{T}$ is generated by the instruction proposer in the MIPRO (Opsahl-Ong et al., 2024). For SMILE training, we set the training sampling budget $B = 500$ and tune $\tau \in \{0.1, 1\}$ and $\beta \in \{10^{-2}, 5 \times 10^{-3}, 10^{-3}\}$ on $\mathcal{D}_{\text{eval}}$. We use a batch size of 64, learning rate $10^{-2}$, and train for 20 epochs.

### 3.2. Main Results

Table 1 reports results on six datasets across two LLM backbones. We draw three conclusions. **(i) SMILE consistently outperforms IO-only, ICL-only, and post-hoc IO+ICL baselines.** SMILE achieves the best average score on both

Qwen3-4B-Instruct (63.0) and Llama3.1-8B-Instruct (60.7), improving over the strongest joint baselines like APE+R.S. (59.8/57.4). The gains are especially pronounced on reasoning datasets like Salient on both backbones, indicating that optimizing prompt components jointly yields more reliable improvements. **(ii) ICL is particularly important for smaller LLMs, while joint optimization is strongest overall.** On Qwen3-4B-Instruct, ICL-only methods outperform IO-only baselines, highlighting the importance of demonstration selection under limited model capacity. Meanwhile, IO+ICL methods outperform both families, suggesting that instructions and demonstrations provide complementary benefits; this hierarchy largely persists on Llama3.1-8B-Instruct and is consistent with the conclusion of (Wan et al., 2024), reinforcing the need to optimize both components. **(iii) Modeling prompt interactions and query-adaptivity explains SMILE's advantage over joint baselines.** Post-hoc combinations implicitly assume that a "best" instruction and a "best" demonstration set compose well, yet Table 1 shows SMILE is consistently stronger than APE+R.S. and MIPRO, particularly on datasets with heterogeneous query-level reasoning needs. This supports our design choices: SMILE accounts for interactions among prompt components and adapts selection to each query (query–instruction and query–demonstration interactions), whereas existing joint baselines typically optimize prompts that are applied uniformly across inputs, which can be suboptimal when *no* single prompt works best for all queries.

*Table 2.* **Component ablations on Qwen3-4B-Instruct.** Performance of SMILE and five variants obtained by removing one expert ($f^{SS}$, $f^{SQ}$, $f^{SI}$), removing the query–instruction scorer ($s_{QI}$), or using softmin aggregation at inference (Agg=softmin). Higher is better.

| Method | GSM8K | GPQA | Financial Phrasebank | XSum | Date | Salient | Avg. |
|---|---|---|---|---|---|---|---|
| **SMILE w/o $f^{SS}$** | 90.8 | 38.4 | 94.0 | 18.5 | 44.8 | 68.2 | 59.1 |
| **SMILE w/o $f^{SQ}$** | 90.4 | 35.8 | 92.0 | 18.4 | 60.8 | 53.6 | 58.5 |
| **SMILE w/o $f^{SI}$** | 90.8 | 38.4 | 93.6 | 17.6 | 61.2 | 69.2 | 61.8 |
| **SMILE w/o $s_{QI}$** | 88.0 | 31.8 | 92.8 | 18.1 | 54.8 | 62.2 | 58.0 |
| **SMILE (Agg=mean)** | 90.4 | 38.4 | 93.2 | 18.4 | 50.8 | 60.0 | 58.5 |
| **SMILE (Agg=softmin)** | 88.4 | 37.4 | 93.6 | 18.3 | 60.0 | 68.4 | 61.0 |
| **SMILE** | **91.2** | **38.9** | **96.0** | **18.5** | **63.2** | **70.0** | **63.0** |

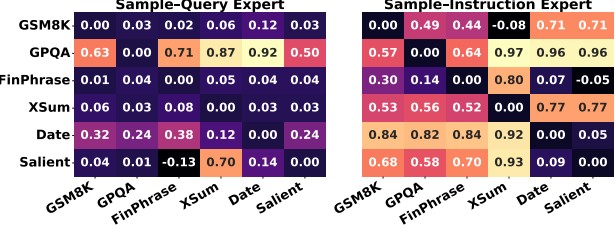

*Figure 3.* **Cross-LLM alignment of expert weight distributions.** Heatmaps visualize the diagonal-centered relative divergence between task-specific models trained with Qwen3-4B-In (rows) vs. Llama3.1-8B-In (columns). Positive off-diagonals indicate stronger cross-LLM agreement for the same task than for mismatched tasks. Left: SQ expert. Right: SI expert.

*Table 3.* **Generalization to Gemini2.5-Flash.** We evaluate Gemini2.5-Flash as the backbone LLM with an ICL budget of $K=10$, comparing baselines to SMILE trained using feedback from Qwen3-4B-In (SMILE-Q) and Llama3.1-8B-In (SMILE-L).

| Method | GSM8K | GPQA | FinPhrase | Date | Salient | Avg. |
|---|---|---|---|---|---|---|
| **Vanilla** | 94.8 | 48.5 | 91.2 | 84.4 | 75.2 | 78.8 |
| **Nearest** | 95.4 | 50.8 | 94.2 | 96.0 | 76.0 | 82.5 |
| **APE+R.S.** | 94.4 | 53.6 | 94.4 | 95.6 | 77.6 | 83.1 |
| **SMILE-Q** | 94.4 | 52.2 | 96.4 | 97.4 | 78.0 | 83.7 |
| **SMILE-L** | 95.2 | 54.8 | 95.2 | 96.6 | 77.2 | **83.8** |

### 3.3. Generalizability of SMILE

We evaluate the generalizability of SMILE from two complementary perspectives: (i) whether the learned surrogate exhibits consistent, task-specific behavior across different LLMs, and (ii) whether a surrogate trained on open-source LLM feedback can transfer to proprietary black-box APIs.

First, we measure cross-LLM alignment of the learned expert-weight distributions. Let $D_{ij}$ denote the Jensen–Shannon divergence between the expert-weight distribution trained on task $i$ using Qwen3-4B-Instruct feedback and the one trained on task $j$ using Llama3.1-8B-Instruct feedback. Figure 3 reports the diagonal-centered relative distance $R_{ij} = (D_{ij} - D_{ii})/D_{ij}$. A positive off-diagonal entry ($R_{ij} > 0$) means that, for task $i$, the surrogate trained with Qwen feedback is closer to the surrogate trained with Llama feedback on the *same* task ($i$) than on a *different* task ($j$). Across both experts ($f^{SQ}$ and $f^{SI}$), the heatmaps show predominantly positive off-diagonal entries, suggesting that

the surrogate learns task-specific weighting patterns that are largely consistent across feedback from different LLMs.

Second, we test transfer to proprietary LLMs by using Gemini2.5-Flash as the backbone LLM. Since API-based models are costly and do not expose internal signals (e.g., token-level log-likelihood), we reuse the surrogates trained with open-source LLM feedback (SMILE-Q / SMILE-L) to guide joint optimization. Table 3 shows that both SMILE-Q and SMILE-L outperform baselines. These results indicate that SMILE provides an effective and cost-efficient optimization strategy for proprietary black-box LLMs. We observe similar results on GPT-5.2 (Appendix A).

### 3.4. Ablation Study of Components

We ablate key components of SMILE on Qwen3-4B-Instruct by removing each expert ($f^{SS}$, $f^{SQ}$, $f^{SI}$), removing the explicit query–instruction scorer ($s_{QI}$), and replacing min aggregation with softmin at inference time (Agg=softmin). From Table 2, removing any component reduces the average score, confirming that the experts and the query–instruction signal are complementary. The largest degradations come from query-aware components: w/o $f^{SQ}$ drops Avg. from 63.0 to 58.5, and w/o $s_{QI}$ further drops to 58.0, consistent with the importance of selecting query-relevant demonstrations for effective ICL (Liu et al., 2022) and with instruction sensitivity varying across queries. While w/o $f^{SS}$ and w/o $f^{SI}$ yield smaller average drops, they are crucial on specific tasks: removing $f^{SS}$ sharply hurts Date (63.2→44.8), suggesting a strong need for diverse/coverage-oriented demonstrations, and removing $f^{SI}$ reduces XSum (18.5→17.6), indicating the value of selecting demonstrations that better align with instruction constraints in summarization. We also observe that SMILE outperforms both softmin and mean aggregation variants. This suggests that the hard min is not merely an arbitrary aggregation choice, but is important for the intended EDSF-style robust conjunction. Intuitively, the three experts encode complementary requirements for a good demonstration set: coverage, query relevance, and instruction compatibility. The min aggregation implements a bottleneck criterion, so a set is not scored highly if it is strong on one axis but weak on another. In contrast, mean aggregation allows a large score from one expert to compen-

sate for a weak score from another, which is less consistent with our goal that all prompt components should be jointly reliable. We train with softmin for optimization stability, but use the hard $\min$ at inference to preserve this bottleneck behavior; smoothing or averaging can dilute the weakest expert, leading to lower average performance. Results on Llama3.1-8B-Instruct are deferred to Appendix A.

## 4. Conclusion

Prompt optimization is increasingly a context engineering problem: effective performance depends on how multiple prompt components (instructions, demonstrations, and the query) interact. Building on the empirical regularity that ICL exhibits diminishing returns, we propose SMILE, an instruction-conditioned EDSF surrogate that explicitly models prompt component interactions while enabling efficient greedy selection. Across diverse tasks and LLM backbones, SMILE delivers consistent gains over baselines. More broadly, our results point to a context-engineering direction: replacing brittle "optimize-then-compose" pipelines with interaction-aware objectives that respect LLM inductive biases (e.g., diminishing returns) while remaining tractable for per-query prompt construction.

## Acknowledgements

This work is supported in part by the National Science Foundation (NSF) under grants CPS-2313110, ECCS-2143559, IIS-2006844, IIS-2144209, IIS-2223769, CNS-2154962, BCS-2228534, and CMMI-2411248; the Office of Naval Research (ONR) under grant N000142412636; and the Commonwealth Cyber Initiative (CCI) under grant VV1Q24-011.

## Impact Statement

This paper advances machine learning by improving prompt optimization for large language models. The proposed method can make LLMs more reliable and sample-efficient across tasks, but any broader societal impacts depend on downstream applications; we do not identify specific consequences that require separate discussion here.

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

# A. Additional Experimental Results

## A.1. Sensitivity to the Number of ICL Examples

We study how performance varies with the ICL budget $K$ on the Salient dataset (Figure 4). We compare SMILE with an ICL baseline (Nearest) and a post-hoc IO+ICL combination (APE+R.S.) under two backbone LLMs, sweeping $K \in \{1, 2, 4, 8, 16, 32\}$.

Two trends are consistent across both LLMs. First, increasing $K$ improves all methods but exhibits diminishing returns: gains are largest from $K{=}1$ to $K{\leq}8$, and performance largely saturates beyond $K{=}16$. Second, SMILE is markedly more *sample-efficient*: it dominates baselines for small budgets and reaches near-saturated performance with substantially fewer demonstrations. For example, on both backbones, SMILE achieves strong performance already at $K{\leq}4$, while Nearest and APE+R.S. require larger $K$ to

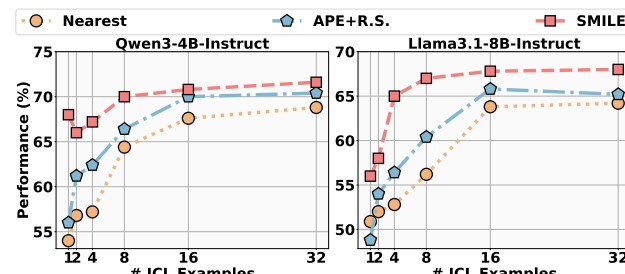

*Figure 4.* **Sensitivity to the ICL budget on Salient.** Performance as a function of the number of in-context examples $K$ selected by different methods (Nearest, APE+R.S., and SMILE) under Qwen3-4B-Instruct and Llama3.1-8B-Instruct.

approach similar levels. This behavior supports our motivation that selecting demonstrations (and instructions) by explicitly modeling their interactions yields better *marginal utility per example*. In practical settings where context length and API cost limit $K$, the advantage of SMILE is therefore most pronounced.

## A.2. Ablation of the Concave Activation

*Table 4.* **Ablation of concave activation** $\phi$. We compare the default $\phi(x) = \log(1+x)$ with a thresholded alternative $\phi(x) = \min\{x, 0.8\}$ on Qwen3-4B-Instruct and Llama3.1-8B-Instruct. Higher is better.

| Method | GSM8K | GPQA | Financial Phrasebank | XSum | Date | Salient | Avg. |
|---|---|---|---|---|---|---|---|
| *Qwen3-4B-Instruct* | | | | | | | |
| SMILE (Act=min) | 90.4 | 38.4 | 93.2 | 18.4 | 50.8 | 60.0 | 58.5 |
| SMILE | **91.2** | **38.9** | **96.0** | **18.5** | **63.2** | **70.0** | **63.0** |
| *Llama3.1-8B-Instruct* | | | | | | | |
| SMILE (Act=min) | 81.2 | 28.2 | 92.6 | 22.8 | 56.0 | 57.6 | 56.4 |
| SMILE | **85.2** | **30.3** | 92.8 | **22.8** | **66.0** | **67.2** | **60.7** |

Our surrogate uses a concave activation $\phi(\cdot)$ to model diminishing returns. While $\phi(x) = \min\{x, \delta\}$ is a valid concave choice, it introduces an additional threshold hyperparameter $\delta$ and becomes piecewise-linear, which can be overly coarse when the underlying gains decay smoothly. We therefore adopt $\phi(x) = \log(1 + x)$ as the default.

To verify this choice, we compare our default surrogate with an alternative using $\phi(x) = \min\{x, 0.8\}$ (motivated by the fact that most channel-wise scores are normalized to $[0, 1]$). Table 4 shows that $\log(1 + x)$ consistently outperforms the min-threshold variant on both backbones (63.0 vs. 58.5 on Qwen3-4B; 60.7 vs. 56.4 on Llama3.1-8B). We attribute the gap to two factors: (i) $\log(1 + x)$ is strictly nonlinear and continuously concave over $[0, 1]$, providing a smoother mapping from aggregated scores to utility and better capturing graded diminishing returns; and (ii) $\min\{x, \delta\}$ is sensitive to the choice of $\delta$. Since $x \in [0, 1]$, setting $\delta$ too large reduces it to an identity mapping, while a smaller $\delta$ requires tuning on a dev set, adding instability and extra selection cost. Overall, $\log(1 + x)$ offers both better accuracy and a more robust, tuning-free design.

## A.3. Ablation Study and Generalizability of SMILE with Alternative Backbones

This section reports complementary results that further support the findings in Sections 3.3 and 3.4.

**Transfer to proprietary LLMs (GPT-5.2).** Table 5 evaluates GPT-5.2 as the backbone LLM. As with Gemini2.5-Flash, we reuse surrogates trained from open-source LLM feedback (SMILE-Q / SMILE-L) to guide joint optimization without access to internal signals. Both SMILE-Q and SMILE-L outperform strong baselines on GPT-5.2, reinforcing that SMILE transfers well and provides a cost-efficient strategy for proprietary black-box models.

**Component ablations on Llama3.1-8B-Instruct.** Table 6 repeats the component ablations on Llama3.1-8B-Instruct by

*Table 5.* **Generalization to GPT-5.2.** We evaluate GPT-5.2 as the backbone LLM with an ICL budget of $K=10$, comparing baselines to SMILE trained using feedback from Qwen3-4B-Instruct (SMILE-Q) and Llama3.1-8B-Instruct (SMILE-L).

| Method | GSM8K | GPQA | FinPhrase | Date | Salient | Avg. |
|---|---|---|---|---|---|---|
| **Vanilla** | 96.4 | 50.0 | 96.2 | 84.4 | 69.2 | 79.2 |
| **Nearest** | 96.8 | 52.0 | 96.4 | 86.8 | 72.4 | 80.9 |
| **APE+R.S.** | 96.8 | 53.5 | 96.4 | 92.6 | 76.4 | 83.1 |
| **SMILE-Q** | 97.2 | 54.5 | 96.8 | 94.4 | 76.0 | 83.8 |
| **SMILE-L** | 96.8 | 53.5 | 98.6 | 96.8 | 78.8 | **84.9** |

*Table 6.* **Component ablations on Llama3.1-8B-Instruct.** Performance of SMILE and five variants obtained by removing one expert ($f^{SS}$, $f^{SQ}$, $f^{SI}$), removing the query–instruction scorer ($s_{QI}$), or using softmin aggregation at inference (Agg=softmin). Higher is better.

| Method | GSM8K | GPQA | Financial Phrasebank | XSum | Date | Salient | Avg. |
|---|---|---|---|---|---|---|---|
| **SMILE w/o $f^{SS}$** | 82.4 | 29.8 | 92.2 | 22.9 | 57.6 | 63.6 | 58.1 |
| **SMILE w/o $f^{SQ}$** | 81.6 | 26.8 | 89.6 | 22.4 | 60.4 | 50.0 | 55.2 |
| **SMILE w/o $f^{SI}$** | 80.8 | 28.8 | 92.6 | 21.4 | 62.4 | 67.2 | 58.8 |
| **SMILE w/o $s_{QI}$** | 82.8 | 22.2 | 91.8 | 19.6 | 59.2 | 60.8 | 56.1 |
| **SMILE (Agg=mean)** | 81.2 | 28.2 | 92.6 | 22.8 | 56.0 | 57.6 | 56.4 |
| **SMILE (Agg=softmin)** | 82.8 | 26.2 | **93.2** | 22.8 | 63.2 | 65.2 | 58.9 |
| **SMILE** | **85.2** | **30.3** | 92.8 | **22.8** | **66.0** | **67.2** | **60.7** |

removing each expert, removing the explicit query–instruction scorer, and replacing $\min$ aggregation with softmin or mean at inference. The largest degradations again come from query-aware components: removing $f^{SQ}$ reduces Avg. from 60.7 to 55.2, and removing $s_{QI}$ reduces Avg. to 56.1. This mirrors the trends observed in Section 3.4 and further highlights the importance of query-adaptive selection.

### A.4. Sensitivity to the instruction candidate pool.

SMILE operates over a predefined candidate instruction set, as is common in candidate-based prompt optimization methods. However, the method is not tied to a particular upstream instruction generator. Its core role is to jointly select an instruction and a query-adaptive demonstration set from the available pool. To examine whether the gains depend on using the MIPRO-generated instruction pool, we additionally construct candidate instruction sets using APE and OPRO, and evaluate SMILE with each pool on Qwen3-4B-Instruct.

Table 7 shows that SMILE remains consistently strong across different instruction generators. In particular, replacing the MIPRO instruction pool with APE or OPRO leads to comparable performance on Financial PhraseBank, Date, and Salient. These results suggest that SMILE's benefit is not primarily due to a specific upstream instruction generator, but rather comes from its joint optimization mechanism that adapts the selected instruction–demonstration pair to each query.

*Table 7.* Sensitivity to instruction candidate pools generated by different methods on Qwen3-4B-Instruct. Higher is better.

| Instruct Pool | FinPhrase | Date | Salient |
|---|---|---|---|
| **w. APE** | 96.4 | 62.8 | 68.8 |
| **w. OPRO** | 94.8 | 64.0 | 70.4 |
| **w. MIPRO** | 96.0 | 63.2 | 70.0 |

### A.5. Sensitivity to the instruction pool size.

We further study how SMILE behaves as the instruction candidate pool grows. Although SMILE operates over a predefined instruction pool, enumerating all candidate instructions can become costly when the pool is large. Therefore, we optionally apply a lightweight prefiltering step in our implementation: for each query $q$, we rank candidate instructions by the query–instruction score $s_{QI}(q, T)$ and keep the top $K_T$ instructions as $\mathcal{T}_q$. SMILE then performs joint instruction–demonstration selection only over $\mathcal{T}_q$.

We conduct a sensitivity study on Salient with Qwen3-4B-Instruct. We vary the full instruction pool size $|\mathcal{T}| \in \{5, 10, 20, 50, 100\}$ and compare two settings: using the full pool $\mathcal{T}$, and using the prefiltered pool $\mathcal{T}_q$ with $K_T = 10$ instructions, or all instructions when $|\mathcal{T}| < 10$. As shown in Table 8, performance improves as the candidate pool grows from 5 to 20 instructions and then largely plateaus. This suggests that larger pools are helpful up to a point, but many

additional instructions become paraphrastic variants rather than genuinely distinct prompting strategies. For example, in the Salient task, we observe generated instructions that differ only in wording while encoding essentially the same task decomposition and output format. One representative pair is:

> **Instruction 1.** "You are a multiple-choice solver. Read the German source sentence and its English translation carefully. Identify the type of translation error (e.g., word substitution, grammatical error, missing word, numerical misrepresentation, factual shift, or named entity error). Do minimal reasoning, then return only the choice the correct answer by selecting one of the options (e.g., '(A)', '(B)')."

> **Instruction 2.** "You are a multiple-choice solver. Read the German source sentence and its English translation carefully. Your task is to identify the type of translation error (e.g., word substitution, grammatical mismatch, numerical inaccuracy, factual shift, or missing entity). Do minimal reasoning, then return only the choice the correct answer by selecting one of the options (e.g., '(A)', '(B)')."

Although these instructions differ slightly in wording, they express the same high-level strategy: inspect the source–translation pair, identify the error type, and output only the multiple-choice option. This helps explain why larger instruction pools improve performance initially but quickly saturate: beyond a moderate pool size, many additional candidates are paraphrastic variants rather than genuinely new prompting strategies.

Prefiltering preserves comparable performance while substantially reducing the number of instruction candidates considered during joint selection. With $K_T = 10$, the prefiltered variant closely tracks the full-pool performance even when $|\mathcal{T}|$ grows to 50 or 100. This indicates a favorable efficiency–quality tradeoff: the query–instruction score can remove many low-priority or redundant instructions before the more expensive instruction-conditioned demonstration selection step. We note that the formal greedy guarantee in

*Table 8.* Sensitivity to instruction pool size on Salient with Qwen3-4B-Instruct.

| **Pool** | 5 | 10 | 20 | 50 | 100 |
|---|---|---|---|---|---|
| $\mathcal{T}$ | 65.2 | 70.0 | 70.8 | 71.2 | 71.2 |
| $\mathcal{T}_q$ | 65.2 | 70.0 | 70.4 | 70.4 | 70.8 |

Section 2.6 applies to demonstration selection for a fixed candidate instruction set; prefiltering is a practical heuristic for reducing the instruction search space and does not change the greedy selection procedure within the retained pool. In many automatic prompt optimization pipelines, the final instruction pool is already moderate; for example, following (Wan et al., 2024), our main experiments use around ten candidate instructions.

# B. Related Work

**Instruction optimization.** Instruction optimization (IO) treats instruction design as a search problem: generate candidate instructions, evaluate them on a held-out set using task metrics or LLM feedback, and iteratively refine the candidate pool under a limited query budget (Shi et al., 2024; Sahoo et al., 2024; Ramnath et al., 2025). Representative approaches span synthesis-and-select pipelines such as APE (Zhou et al., 2022), optimizer-in-the-loop methods such as OPRO (Yang et al., 2024), Monte Carlo sampling such as ProTeGi (Pryzant et al., 2023) and PromptAgent (Wang et al., 2023), and population/evolutionary strategies such as PromptBreeder (Fernando et al., 2024); other lines explore gradient- or RL-inspired prompt optimization for discrete text prompts (Shin et al., 2020; Deng et al., 2022; Yuksekgonul et al., 2024). More recently, GEPA (Agrawal et al., 2025) extends instruction optimization to compound LLM systems by evolving prompts via natural-language reflection over system trajectories, achieving strong improvements with substantially fewer rollouts than RL-based adaptation, such as GRPO (Shao et al., 2024). Nevertheless, GEPA can still require thousands of rollouts, which becomes costly for long prompts and API-based models. More broadly, most IO approaches optimize a small set of global instructions (often only a single instruction) intended to transfer across inputs, and when demonstrations are used, they are often optimized separately or treated as add-ons. In contrast, we view prompting as *context engineering*: instructions and demonstrations interact nontrivially, and the best instruction–demonstration pair can be query-dependent. SMILE explicitly models these interactions to jointly optimize instructions and ICL demonstrations.

**In-context learning.** In-context learning (ICL) enables large language models (LLMs) to perform tasks by conditioning on a small set of input–output demonstrations (Brown et al., 2020). It has shown strong empirical success across a wide range of settings, including summarization (Jain et al., 2023; Baek et al., 2025) and reasoning (Lee et al., 2024; Chen et al., 2024b), among many others (Agrawal et al., 2022; Sia & Duh, 2023; Wang et al., 2021; He et al., 2023). To effectively harness ICL, prior work has proposed adaptive strategies for selecting demonstrations from a large pool (Rubin et al., 2022; Ye et al., 2023; Chen et al., 2025b). This selection is inherently combinatorial, and the search space grows rapidly with the ICL budget (Wang et al., 2024b). To make the problem tractable, many methods rely on heuristic objectives, often cast

as submodular functions based on semantic similarity (Liu et al., 2021) or diversity/coverage (Levy et al., 2022; Kumari et al., 2024), which enable efficient greedy selection with a $(1 - 1/e)$ approximation guarantee under the assumed objective. However, these objectives are typically hand-designed proxies: they presume that the proxy score correlates with true ICL utility, yet recent analyses show that ICL gains can depend sensitively on prompt structure and may deviate from such heuristics (Agarwal et al., 2024; Bertsch et al., 2025; Baek et al., 2025). As a result, these proxies can be misaligned with what the LLM actually rewards.

**Joint IO and ICL optimization.**   Although instruction optimization (IO) and in-context learning (ICL) share the goal of improving task performance via prompting, they have largely been studied in isolation. Wan et al. (2024) systematically compare IO and ICL methods, as well as their post-hoc combinations, and show that the two are complementary: combining optimized instructions with optimized demonstrations can yield gains beyond either component alone. Fewer works explicitly *jointly* optimize both instruction and demonstrations. EASE (Wu et al., 2024) searches over prompt structures by selecting instruction–example combinations from a predefined candidate pool using bandit-based optimization. Adv-ICL (Do et al., 2024) formulates prompt optimization as a minimax game, where LLMs generate, critique, and iteratively edit prompts (instructions plus exemplars) via adversarial feedback. MIPRO (Opsahl-Ong et al., 2024) targets multi-stage LM programs and improves both instructions and demonstrations through program-aware proposal, bootstrapped demos, and stochastic mini-batch evaluation for credit assignment under sparse end-task feedback. Finally, Wang et al. (2024a) adopt a mixture-of-experts view by partitioning the problem space into regions, each governed by an expert prompt (instruction + demos), and route each query to its closest expert at inference. A common limitation of existing joint approaches is that they primarily optimize *global* prompts (a single instruction or a fixed instruction–demonstration set) intended to transfer across inputs, and thus do not explicitly model query-specific interactions between the instruction, demonstrations, and the test query. This can be suboptimal when no universal prompt works best across heterogeneous queries.

## C. Baselines

- **Vanilla:** We query the LLM zero-shot using a fixed seed instruction (no optimization and no demonstrations). The seed instructions for each task are provided in Appendix G

- **APE (Zhou et al., 2022):** APE is a canonical LLM-based instruction optimizer. At each iteration, it evaluates a population of candidate instructions on the validation set $\mathcal{D}_{\text{dev}}$, then prompts an optimizer LLM to generate new candidates by paraphrasing the top-performing instructions. This loop repeats until convergence. Following the customized prompting setup of Wan et al. (2024), we initialize APE from the default seed instruction (rather than instruction induction from exemplars (Honovich et al., 2023)).

- **OPRO (Yang et al., 2024):** OPRO improves instructions via implicit optimization. Starting from the seed instruction, each iteration provides the optimizer model with previously tried instructions and their validation scores, and asks it to produce a better instruction. Unlike APE, OPRO does not explicitly request paraphrasing or reflection, treating the optimizer LLM as a black-box optimizer and simply asking LLM to "come up with a better instruction".

- **Random (Wan et al., 2024):** Given an ICL budget $K$, we uniformly sample $K$ input–output demonstrations $\mathcal{S} \subset \mathcal{D}$ and prepend them to the Vanilla prompt. The model then generates the final prediction (including any intermediate reasoning it produces).

- **Diversity (Wan et al., 2024):** To encourage diverse demonstrations, we cluster the pool $\mathcal{D}$ into $K$ clusters using $k$-means over embeddings from Qwen3-Embedding-0.6B (Zhang et al., 2025), and select the example closest to each centroid to form $\mathcal{S}$ (cf. Zhang et al. (2022)).

- **Nearest (Wan et al., 2024):** We retrieve demonstrations most similar to the current test input by cosine similarity in an embedding space (Liu et al., 2022). Concretely, we compute embeddings with Qwen3-Embedding-0.6B (Zhang et al., 2025) and select the top-$K$ demonstrations to form $\mathcal{S}$.

- **APE+R.S. (Wan et al., 2024):** R.S. denotes *random search*. Following the ICL selection procedure in DSPy (Khattab et al., 2023), we randomly sample $m$ candidate demonstration sets of size $K$, i.e., $\{\mathcal{S}_1, \mathcal{S}_2, \ldots, \mathcal{S}_m\}$, where each $\mathcal{S}_\ell = \{e_i^\ell\}_{i=1}^K$ for $\ell \in [m]$. We evaluate each candidate set on the held-out validation set $\mathcal{D}_{\text{dev}}$ and choose the best-performing $\mathcal{S}^*$ as the final ICL demonstration set. For the instruction, we use the best instruction produced by APE (Zhou et al., 2022). Overall, APE+R.S. is a representative *post hoc* combination that optimizes the instruction

and demonstrations separately, implicitly assuming that strong instructions and strong demonstrations compose. As reported in Wan et al. (2024), this baseline provides a strong performance–cost trade-off in terms of LLM calls.

- **MIPRO (Opsahl-Ong et al., 2024):** MIPRO jointly optimizes both instructions and few-shot demonstrations. It first proposes a candidate set of instructions for each module and bootstraps candidate demonstration sets, then uses a Bayesian surrogate model to perform credit assignment and search over combinations of these candidates. The best candidates are periodically re-evaluated on the full train set, and the highest-scoring configuration is returned. We adapt the official DSPy implementation (Khattab et al., 2023) and set `max_labeled_demos` to $K$.

## D. Datasets

- **GSM8K (Cobbe et al., 2021):** GSM8K is a dataset of high-quality grade school math word problems. The dataset was created to support the task of question answering on basic mathematical problems that require multi-step reasoning. We use it to evaluate mathematical problem-solving. An example question is: "Adam has an orchard. Every day for 30 days he picks 4 apples from his orchard. After a month, Adam has collected all the remaining apples, which were 230. How many apples in total has Adam collected from his orchard?"

- **GPQA (Rein et al., 2023):** GPQA is a multiple-choice question answering benchmark with challenging graduate-level questions in biology, physics, and chemistry, designed to test advanced scientific reasoning. We use the Diamond split as the test set, and partition the remaining data into $\mathcal{D}$ and $\mathcal{D}_{\text{val}}$. An example question is: "Cas9 nuclease and the restriction enzyme EcoRI are produced by the bacterial system. Which of the following statements are true regarding both enzymes? I. They are both endonucleases that perform cleavage activity at the specific nucleotide sequence II. They both create double-strand breaks in DNA. III. Both Cas9 and EcoRI cut double-stranded DNA forming overhangs in the DNA. IV. They are both part of bacterial defense mechanisms against foreign DNA Choices: (A) I, II, III (B) I, II, III, IV (C) I, III, IV (D) I, II, IV"

- **Financial PhraseBank (Malo et al., 2014):** Financial PhraseBank is a sentiment analysis dataset consisting of sentences from English financial news, categorized by sentiment. The annotators were instructed to label each sentence as *positive*, *negative*, or *neutral* from an investor perspective. We construct our dataset using the `sentences_allagree` subset, which contains instances with 100% annotator agreement. An example question is: "Kesko Agro Eesti, the retailer and wholesaler of grain, agricultural and warehousing machinery and accessories, had net sales of 81 million euros in 2007, an increase by one-tenth over the preceding year."

- **XSum (Narayan et al., 2018):** Extreme Summarization (XSum) is an abstractive single-document summarization benchmark of BBC news articles. The goal is to generate a concise one-sentence summary answering the question, "What is the article about?"

- **Date (Suzgun et al., 2023):** This dataset is designed for the date understanding task, where models answer a provided question based on a small set of sentences related to a particular date. An example question is: "On May 9th, 2017 Jane bought 40 eggs. She ate one per day. Today she ran out of eggs. What is the date one week ago from today in MM/DD/YYYY? Options: (A) 06/11/2017 (B) 06/11/1923 (C) 07/02/2017 (D) 07/11/2017 (E) 11/11/2016 (F) 06/07/2017"

- **Salient (Suzgun et al., 2023):** This dataset is designed for translation error detection, where, given a German source sentence and its English translation, the task is to determine the salient error type in the translation. An example question is: "Source: Ricoldo da Monte di Croce war ein Orientmissionar. Translation: Ricoldo was an Orient missionary. The translation contains an error pertaining to: (A) Modifiers or Adjectives (B) Numerical Values (C) Negation or Antonyms (D) Named Entities (E) Dropped Content (F) Facts"

## E. Theoretical Proofs.

**Lemma E.1** (Good step condition under submodularity). *Let $f : 2^{\mathcal{D}} \to \mathbb{R}_+$ be a monotone submodular function, and let $f^{\star} := \max_{\mathcal{A} \subseteq \mathcal{D}, \, |\mathcal{A}| \leq k} f(\mathcal{A})$. For any partial solution $\mathcal{A} \subseteq \mathcal{D}$ with $|\mathcal{A}| < k$, there exists an element $x \in \mathcal{D} \setminus \mathcal{A}$ such that*

$$f(\mathcal{A} \cup \{x\}) - f(\mathcal{A}) \;\geq\; \tfrac{1}{k} \left( f^{\star} - f(\mathcal{A}) \right). \tag{10}$$

*Proof.* Let $\mathcal{A}^\star \subseteq \mathcal{D}$ be an optimal solution of size at most $k$, so that $f(\mathcal{A}^\star) = f^\star$. By monotonicity,

$$f^\star - f(\mathcal{A}) \leq f(\mathcal{A} \cup \mathcal{A}^\star) - f(\mathcal{A}).$$

By submodularity,

$$f(\mathcal{A} \cup \mathcal{A}^\star) - f(\mathcal{A}) \leq \sum_{x \in \mathcal{A}^\star \setminus \mathcal{A}} \big(f(\mathcal{A} \cup \{x\}) - f(\mathcal{A})\big).$$

Since the right-hand side is a sum of at most $k$ terms, there must exist some element $x \in \mathcal{A}^\star \setminus \mathcal{A}$ with marginal contribution at least $\frac{1}{k}(f^\star - f(\mathcal{A}))$, which is exactly Equation 8. □

**Theorem E.2** (High-probability guarantee for greedy under per-step distortion). *Let $f : 2^{\mathcal{D}} \to \mathbb{R}_+$ be a monotone set function, and let $f^\star := \max_{\mathcal{A} \subseteq \mathcal{D}, |\mathcal{A}| \leq k} f(\mathcal{A})$. Consider the greedy algorithm, which constructs a sequence of sets $\mathcal{A}_0 = \varnothing, \mathcal{A}_1, \ldots, \mathcal{A}_k$ by iteratively adding one element at a time. Suppose that at each iteration $t \in \{1, \ldots, k\}$, with probability at least $1 - \varepsilon$ (possibly conditional on the past history), the "good step" condition Equation 8 holds. Let $\delta \in (0, 1)$. Then, with probability at least $1 - \delta$, the greedy solution $\mathcal{A}_k$ satisfies*

$$\frac{f(\mathcal{A}_k)}{f^\star} \geq 1 - \exp\left(-1 + \varepsilon + \sqrt{2\ln(1/\delta)/k}\,\right). \tag{11}$$

*Proof.* Define $GAP_t := f^\star - f(\mathcal{A}_t)$. If step $t$ is "good", condition Equation 8 implies that after the greedy selection

$$GAP_t \leq \left(1 - \tfrac{1}{k}\right) GAP_{t-1}.$$

If step $t$ is "bad", monotonicity implies $GAP_t \leq GAP_{t-1}$.

For each step $t$, define the indicator variable

$$H_t := \begin{cases} 1, & \text{if } \max_{x \notin \mathcal{A}_{t-1}} \big(f(\mathcal{A}_{t-1} \cup \{x\}) - f(\mathcal{A}_{t-1})\big) \geq \tfrac{1}{k}\big(f^\star - f(\mathcal{A}_{t-1})\big), \\ 0, & \text{otherwise.} \end{cases}$$

Thus $H_t \in \{0, 1\}$ indicates whether the "good step" condition (Equation 8) is satisfied at iteration $t$. Thus, we can unify the "good" and "bad" steps as

$$GAP_t \leq \left(1 - \tfrac{H_t}{k}\right) GAP_{t-1}.$$

Iterating over $t = 1, \ldots, k$ gives

$$GAP_k \leq \left(\prod_{t=1}^{k}\left(1 - \tfrac{H_t}{k}\right)\right) f^\star \leq \exp\left(-\tfrac{1}{k}\sum_{t=1}^{k} H_t\right) f^\star,$$

where the last inequality uses $1 - u \leq \exp(-u)$. Equivalently,

$$\frac{f(\mathcal{A}_k)}{f^\star} = 1 - \frac{GAP_k}{f^\star} \geq 1 - \exp\left(-\tfrac{1}{k}\sum_{t=1}^{k} H_t\right). \tag{12}$$

It remains to bound $\sum_{t=1}^{k} H_t$. By assumption, $\mathbb{E}[H_t \mid \mathcal{F}_{t-1}] \geq 1 - \varepsilon$ for all $t$, where $\mathcal{F}_{t-1}$ denotes the history up to time $t - 1$. Define a martingale difference sequence

$$M_t = \sum_{i=1}^{t}\left(H_i - \mathbb{E}[H_i \mid \mathcal{F}_{i-1}]\right).$$

We have $|M_t - M_{t-1}| \leq 1$ for all $t$. By Azuma–Hoeffding's inequality, for any $s > 0$,

$$\Pr\big(M_k \leq -s\big) \leq \exp\left(-\tfrac{s^2}{2k}\right).$$

Therefore, with probability at least $1 - \delta$,

$$M_k \geq -\sqrt{2k \ln(1/\delta)}.$$

Combining this with the expectation lower bound gives

$$\sum_{t=1}^{k} H_t = \sum_{t=1}^{k} \mathbb{E}[H_t \mid \mathcal{F}_{t-1}] + M_k \geq (1-\varepsilon)k - \sqrt{2k \ln(1/\delta)}.$$

Plugging this bound into Equation 12, we conclude that with probability at least $1 - \delta$,

$$\frac{f(\mathcal{A}_k)}{f^\star} \geq 1 - \exp\left(-1 + \varepsilon + \sqrt{2\ln(1/\delta)/k}\right).$$

$\square$

*Remark* E.3. Lemma 2.4 shows that the "good step" condition is always satisfied when $f$ is monotone submodular. In this case, $\varepsilon = 0$ and Theorem 2.5 recovers the classical $(1 - 1/e)$ approximation guarantee of Nemhauser et al. (1978). More generally, when $\varepsilon > 0$ but small, the theorem shows that greedy still achieves a near-optimal solution with high probability, up to an additive loss of order $O(\varepsilon)$ and a vanishing concentration term $O(\sqrt{\ln(1/\delta)/k})$.

*Remark* E.4 (Expectation bound). A simpler but coarser argument can be obtained by taking expectations directly. Recall that our analysis of greedy relies on the recursive inequality

$$\frac{f(\mathcal{A}_k)}{f^\star} \geq 1 - \exp\left(-\tfrac{1}{k} \sum_{t=1}^{k} H_t\right).$$

In the fully submodular case (i.e. $H_t = 1$ for all $t$) this recovers the classical $(1 - 1/e)$ ratio. More generally, if with probability at least $1 - \varepsilon$ all steps satisfy the submodularity condition simultaneously, then with the remaining probability we only use the trivial lower bound $0$. Taking expectations gives

$$\mathbb{E}\left[\frac{f(\mathcal{A}_k)}{f^\star}\right] \geq (1-\varepsilon)(1 - 1/e).$$

Thus, the expected approximation ratio also degrades gracefully with $\varepsilon$. Compared to the high-probability version in Theorem 2.5, this argument treats all violations as a single global failure event, and hence yields a weaker but more direct guarantee.

## F. Limitations

While SMILE consistently improves prompt optimization across tasks and models, it has several limitations that are likely beyond the scope of this work.

**When diminishing-returns assumptions may not hold.** SMILE relies on a submodular (diminishing-returns) surrogate to guide selection. This inductive bias can be misaligned with tasks where the utility of an additional demonstration is *non-diminishing* or even *super-additive*, e.g., multi-step reasoning that requires composing multiple complementary facts or rules, problems with strong cross-example dependencies ("you need both examples A and B for the final step"), or tasks where a late example triggers a qualitatively different solution strategy. In such cases, a fixed-budget greedy construction may miss globally optimal combinations.

**Sensitivity to prompt format and instruction heterogeneity.** Although SMILE explicitly models query–instruction and query–demonstration interactions, it still optimizes within a restricted prompt template family. If a task benefits from more radical prompt restructuring (e.g., multi-stage prompting, tool use, self-consistency, or structured scratchpads), the gains from selecting instructions/demonstrations alone may be limited. Relatedly, if the instruction pool contains many near-duplicates or noisy/underspecified instructions, the method can become less stable and may overfit to spurious phrasing patterns.

**Limited guarantees under black-box and distribution shift.** For proprietary black-box LLMs, we reuse surrogates trained from open-source feedback. While we observe good transfer empirically, there is no guarantee that the learned expert-weight distributions remain calibrated under distribution shift (new tasks, new domains, or new API model versions). Handling continual model updates or rapidly changing deployment distributions would require online adaptation and monitoring, which we leave for future work.

# G. Prompts

We provide the prompts used in our experiments for math (GSM8K), question answering (GPQA), sentiment analysis (Financial Phrasebank), summarization (XSum), and reasoning (Date and Salient) tasks in Table 9.

*Table 9.* Prompts used in our experiments.

| Types | Prompts |
|---|---|
| Math | Let's think step by step. Return the final integer in the form ⟦answer⟧ on the last line.

(provide examples here with the following format.)
Question: `<question>`
Answer: `<answer>`

Now, solve the following task.

Sentence: {TARGET_QUERY} |
| Sentiment Analysis | Classify the sentiment as one of: positive, neutral, negative. Answer with exactly one word.

(provide examples here with the following format.)
Sentence: `<sentence>`
Label: `<label>`

Now, solve the following task.

Sentence: {TARGET_QUERY} |
| Summarization | Write a single-sentence abstractive summary that captures the main point. Return exactly one sentence.

(provide examples here with the following format.)
Article: `<article>`
Summary: `<summary>`

Now, solve the following task.

Article: {TARGET_QUERY} |
| Question Answering

or

Reasoning | Answer the multiple-choice question. Do minimal reasoning, then return the choice of the correct answer by selecting one of the options (e.g., '(A)', '(B)').

(provide examples here with the following format.)
Question: `<question>`
Answer: `<answer>`

Now, solve the following task.

Question: {TARGET_QUERY} |

