# OpenReview forum: "SMILE: Extended Deep Submodular Function-Based Instruction and In-context Learning Demonstration Selection"
_ICML.cc/2026/Conference — ICML 2026 regular_

### Official Review · Reviewer_VyvB · 2026-02-19

**Soundness:** 3
**Presentation:** 2
**Significance:** 2
**Originality:** 3
**Overall Recommendation:** 4
**Confidence:** 4

**Summary:**

The authors claim to consider the central issue that instruction optimization (IO) and in-context learning (ICL) demonstration selection are typically optimized separately and combined post hoc, assuming that "best" instructions and "best" demonstrations compose well. They outline an important area of jointly optimizing these interacting prompt components. The authors propose SMILE, which learns an instruction-conditioned Extended Deep Submodular Function (EDSF) surrogate that captures three types of interactions (sample-sample coverage, sample-query relevance, and sample-instruction compatibility). By leveraging the diminishing-returns structure of ICL, SMILE enables efficient greedy selection of instruction-demonstration pairs. Experiments on six datasets and multiple LLM backbones show consistent improvements over IO-only, ICL-only, and existing joint baselines.

**Compliance With Llm Reviewing Policy:**

Affirmed.

**Final Justification:**

The authors have addressed my questions.

**Key Questions For Authors:**

See Weaknesses

**Limitations:**

Yes

**Strengths And Weaknesses:**

Strength:

+ The paper clearly identifies the limitation of decoupled IO+ICL pipelines and provides empirical evidence (Figure 1) that ICL exhibits consistent diminishing returns across instructions, motivating the submodular surrogate approach.
+ Consistent gains across diverse tasks (math, QA, sentiment, summarization, reasoning) and model scales, including effective transfer to black-box proprietary APIs without retraining.

Weakness:

+  The method relies heavily on the diminishing-returns assumption, but tasks with super-additive example interactions (e.g., multi-step reasoning requiring complementary facts) are not deeply analyzed.
+ What is the sensitivity to the instruction pool size |T|? Would the method scale to hundreds of candidate instructions, and how would the prefiltering heuristic affect optimality guarantees?

---

> ### Author Rebuttal · Authors · 2026-03-29
>
> >**W1.** The method relies heavily on the diminishing-returns assumption, but tasks with super-additive example interactions (e.g., multi-step reasoning requiring complementary facts) are not deeply analyzed.
> >
> **Response**: Thank you for raising this point. We agree that tasks with strongly complementary demonstrations can violate the diminishing-returns bias underlying our surrogate. In fact, we **explicitly discuss this in Appendix F**, where we note that SMILE may be less well aligned with settings such as multi-step reasoning or cases where a later example induces a qualitatively different strategy. We emphasize that we do not claim to handle such settings in the current paper. A possible extension would be to introduce additional modules and richer compatibility terms to better capture these more complex interactions, and we'll leave this as an important future work.
>
> However, we would emphasize that this **does not undermine the main contribution** of the paper.  Our goal is not to claim that diminishing returns universally governs all ICL behaviors, but to show that **it provides a principled and effective bias for a broad and practically important class of prompt-optimization problems**. Our contribution is to identify, formalize, and leverage this useful structural bias that is both theoretically grounded and empirically effective. Importantly, the empirical results suggest that this bias is **far from narrow**: SMILE performs well across a heterogeneous set of problems, including math reasoning (GSM8K), scientific QA (GPQA), sentiment analysis (FP), summarization (XSum), and date reasoning (Date). This breadth indicates that diminishing returns is not merely a restrictive assumption, but a useful modeling principle in many realistic settings.
>
> We will revise the discussion to make the scope boundary clearer: SMILE is designed for settings where modular interactions are informative, and we do not claim optimality for tasks dominated by super-additive example interactions. We agree that analyzing or extending the method for such regimes is an important direction for future work.
>
>
> >**W2.** What is the sensitivity to the instruction pool size |T|? Would the method scale to hundreds of candidate instructions, and how would the prefiltering heuristic affect optimality guarantees?
> >
> **Response**: Thank you for this helpful question. We agree that sensitivity to the instruction pool size and the effect of prefiltering are important practical considerations.  We conduct a sensitivity study on Salient with Qwen3-4B as the backbone LLM. We vary the full candidate pool size $|T|\in{5,10,20,50,100}$, and for each pool we form the prefiltered set $T_q$ by keeping the top $K_T=10$ instructions (or all when $|T|<10$).
>
> |  | 5 | 10 | 20 | 50 | 100 |
> |---|---:|---:|---:|---:|---:|
> | $\|T\|$| 65.2 | 70.0 | 70.8 | 71.2 | 71.2 |
> | $\|T_q$\|  | 65.2 | 70.0 | 70.4 | 70.4 | 70.8 |
>
> As $|T|$ increases, performance improves initially and then plateaus. With prefiltering, the same trend holds for $|T_q|$. This suggests that beyond a moderate pool size, many extra instructions are paraphrastic variants rather than genuinely new prompting strategies. This suggests that beyond a moderate pool size, many additional instructions are paraphrastic variants rather than genuinely new prompting strategies. For example, among the generated instructions on the Salient dataset, we observed pairs such as:
>
>
> "You are a multiple-choice solver. Read the German source sentence and its English translation carefully. Identify the type of translation error (e.g., word substitution, grammatical error, missing word, numerical misrepresentation, factual shift, or named entity error). Do minimal reasoning, then return only the choice the correct answer by selecting one of the options (e.g., '(A)', '(B)').",
>
> and
>
> "You are a multiple-choice solver. Read the German source sentence and its English translation carefully. Your task is to identify the type of translation error (e.g., word substitution, grammatical mismatch, missing word, numerical inaccuracy, factual shift, or missing entity). Do minimal reasoning, then return only the choice the correct answer by selecting one of the options (e.g., '(A)', '(B)').",
>
>
> These instructions **differ slightly in wording**, but they encode essentially the same task decomposition. Accordingly, larger instruction pools are helpful up to a point, but their marginal benefit quickly saturates. At the same time, **prefiltering preserves comparable performance while substantially reducing the search cost**, indicating a favorable efficiency–quality tradeoff. In addition, in many practical prompt-optimization pipelines the final **instruction pool is already moderate**; in our implementation, following [1], the typical scale is around 10 candidate instructions.
>
> [1] Wan, X., Sun, R., Nakhost, H., and Arik, S. Teach better or show smarter? on instructions and exemplars in automatic prompt optimization. NeurIPS 2024.

---

> > ### Author Rebuttal · Reviewer_VyvB · 2026-04-03
> >
> > Thank you for your response. I'll increase my score.

---

> > > ### Author Response · Authors · 2026-04-03
> > >
> > > We thank you sincerely for your careful reading and thoughtful feedback.  We are grateful that the additional analyses and clarifications were useful in addressing your concerns.

---

### Official Review · Reviewer_A8CB · 2026-03-11

**Soundness:** 3
**Presentation:** 3
**Significance:** 3
**Originality:** 2
**Overall Recommendation:** 4
**Confidence:** 3

**Summary:**

SMILE jointly selects instructions and in-context learning (ICL) demonstrations, motivated by the observation that optimizing them separately and combining post hoc is brittle because the two interact. The paper observes that ICL performance exhibits diminishing returns that are consistent across instructions, which motivates a submodular surrogate. Specifically, SMILE instantiates an Extended Deep Submodular Function (EDSF)—a pointwise minimum of Deep Submodular Functions—over three expert modules: sample–sample coverage ($f\_{SS}$), sample–query relevance ($f\_{SQ}$), and sample–instruction compatibility ($f\_{SI}$). Each expert is a concave-over-modular DSF; the three are aggregated by hard min. Lightweight scalar calibration weights are learned from pairwise LLM feedback via a ranking loss. At inference, greedy selection produces a demonstration set per candidate instruction, then the best pair is chosen. Theorem 2.5 shows greedy retains near-$(1 - 1/e)$ guarantees when the good-step condition holds with high probability. Experiments across six datasets, two open-source LLMs, and two proprietary APIs show consistent gains over IO-only, ICL-only, and joint baselines.

**Compliance With Llm Reviewing Policy:**

Affirmed.

**Final Justification:**

My concerns were fully addressed in the prior round: the IG channel is validated by Spearman correlations (0.73 XSum, 0.71 Date), the EDSF expressiveness argument is appropriately reframed, and the remaining limitations are honestly acknowledged. I maintain my score of 4: the paper is a solid contribution, but the gains over baselines are incremental, the EDSF framework is borrowed, and the learned parameters are limited to scalar weights.

**Key Questions For Authors:**

1. **How does the IG channel relate to actual ICL interaction?** $IG(e\_i; T)$ measures zero-shot instruction utility on $e\_i$, not the value of $e\_i$ as a demonstration under $T$. Can you provide evidence (e.g., correlation analysis) that high zero-shot IG implies an example is a useful demonstration under that instruction? An ablation isolating IG vs. prototype within $f\_{SI}$ would also help.

2. **Can you report confidence intervals for the main results?** With 250 test instances, a 2% accuracy difference is ~5 examples. Bootstrap CIs or paired tests would clarify which improvements in Table 1 are robust—especially for datasets where SMILE's edge is small or negative.

3. **How does SMILE compare to a weighted sum instead of min?** A linear combination of the same three DSF experts would test whether the EDSF bottleneck structure contributes beyond simply having three complementary scores.

**Limitations:**

The limitations in Appendix F are honest—failure modes of the submodularity assumption, prompt format sensitivity, and lack of transfer guarantees. Not acknowledged: the gap between the EDSF expressiveness theorem and the actual parameterization, the zero-shot nature of the IG signal, and the dependency on MIPRO for instruction generation.

**Strengths And Weaknesses:**

### Strengths

- **Well-motivated problem:** Instruction optimization and demonstration selection are typically studied in isolation, but they interact in query-dependent ways. Framing this as a joint selection problem under a shared surrogate is a natural and useful perspective. The paper explains clearly *why* separate-then-compose is unreliable.

- **Modular, interpretable design:** The three-expert decomposition is clean, and the concave-over-modular template gives the method uniform structure. Learning only scalar weights keeps the method lightweight. The ablations are well-designed and informative—they reveal that different experts matter on different tasks ($f\_{SS}$ critical for Date, $f\_{SQ}$ most important on average), validating the multi-expert setup.

- **Cross-LLM transfer:** Training the surrogate on Qwen or Llama feedback and applying it to Gemini or GPT-5.2 with no additional LLM calls is the use case that matters for API-based models. The Jensen–Shannon heatmaps (Figure 3) provide supporting evidence that task-specific patterns are more stable across LLMs than across tasks.

---

### Weaknesses

- **The expressiveness argument doesn't support the specific instantiation:** Theorem 2.3 establishes that the EDSF *family* can represent any monotone submodular function. But this is about the closure of all possible EDSFs (potentially requiring unbounded $r$ and arbitrary modular channels). SMILE uses exactly $r = 3$ DSFs with 6 hand-designed channels and learns only scalar weights. The gap between "the family is universal" and "this specific parameterization approximates the true ICL utility" is large and unaddressed—one wouldn't cite the universal approximation theorem to justify a specific 5-neuron network. The burden for justifying the surrogate falls entirely on the empirical results, which the paper should acknowledge more honestly.

- **The information gain channel doesn't capture what it claims:** The sample–instruction expert is motivated as capturing "the interaction between demonstrations and the instruction." But $IG(e\_i; T) = \log p(y\_i | T, x\_i) - \log p(y\_i | T\_0, x\_i)$ is a *zero-shot* quantity—it measures how much $T$ helps the model on $e\_i$ without any demonstrations present. This is not the same as whether $e\_i$ is a useful demonstration under $T$ when other examples occupy the prompt. The paper conflates "instruction helps the model on this example" with "this example is a good demonstration under this instruction." The prototype channel inherits the same issue since it's derived from IG-weighted embeddings.

- **Evaluation lacks statistical rigor:** Test sets are capped at 250 instances. No confidence intervals or significance tests are reported. Some differences are small enough to be noise: SMILE ties or loses on XSum for both LLMs and loses on Financial PhraseBank with Llama (92.8 vs. 94.8 for APE+R.S.). The average gains are driven primarily by Date and Salient; without error bars, it's hard to tell which individual improvements are meaningful.

- **White-box dependency is underplayed:** The IG channel requires teacher-forced log-likelihoods, unavailable for black-box APIs. The transfer experiments use surrogates calibrated to a *different* model. The paper presents black-box applicability as a strength, but the method always needs a white-box proxy for training—an important constraint that deserves more explicit discussion.

---

### Minor

- Figure 1 shows diminishing returns on only two datasets. If this is the core motivating observation, all six would be more convincing.
- The instruction pool $\mathcal{T}$ is generated by MIPRO's proposer, making the comparison with MIPRO asymmetric (it's about selection strategy, not the full pipeline).
- Theorem 2.5 is a direct Azuma–Hoeffding application; similar approximate-submodularity guarantees exist (e.g., Das & Kempe 2011) and aren't discussed.

---

> ### Author Rebuttal · Authors · 2026-03-29
>
> >**W1.** Theorem 2.3.
> >
> **Response**: Thank you for this point. Our use of the theorem is more modest: to motivate EDSFs as a reasonable surrogate family for utilities with diminishing returns, consistent with the empirical ICL behavior. More broadly, as noted in the paper, our surrogate is **not** restricted to the particular channels used in the current instantiation. SMILE can incorporate other fixed or learned modular components. We will revise the paper to make it clearer: the theorem motivates the surrogate family, while the value of the concrete SMILE design is supported by the ablations and end-task improvements. (Minor: SMILE  has k(SS)+2(SQ)+2(SI) channels rather 5/6.)
>
> >**W2&Q1.** Regarding IG.
> >
> **Response**: We agree that IG is **not** the same as the true marginal ICL value. Our intent is therefore **not** to present it as an exact estimator, but rather as a lightweight **instruction-compatibility prior**: examples for which T already increases the model’s preference for the correct output are more likely to be coherent/reliable samples under that T, while low or negative-IG examples are less likely to be so. The prototype term is intended to smooth this noisy per-example signal by capturing the region of examples to which the instruction appears most applicable.
>
> To test whether IG is aligned with ICL utility, we sample 50 examples and for each, estimate its **empirical single-demo utility** on test queries as
> $$ U(e_i;T)=E_{q} [r(T,e_i,q)-r(T,q)].$$
> We compute the Spearman correlation between $IG(e_i;T)$ and $U(e_i;T)$. We find strong positive correlations: 0.73 on XSum and 0.71 on Date with Qwen3. This supports using IG as a **meaningful proxy** for instruction-conditioned example utility.
>
> We also ablate the components of fSI. The results suggest that both channels are useful and that combining them is consistently better than using either one alone.
>
> |  | XSum | Date |
> |-|-|-|
> |w/o fSI |17.6| 61.2|
> | IG-only| 18.3|62.4 |
> | Proto-only  | 18.2|62.8 |
> |SMILE |18.5|63.2|
>
> We will revise the paper to make the interpretation more precise.
>
>
> >**W3.** Statistical rigor.
> >
> **Response**:  Thank you for raising this point. We agree that the current version would be stronger with uncertainty estimates. Due to rebuttal-time constraints, we prioritize the two most concerned LLaMA settings: FinPhrase and XSum, and compare with the strongest baseline. Over 5 random seeds, we obtain:
>
> - **FinPhrase:** APE+R.S. **94.4 ± 0.57**, SMILE **93.2 ± 0.33**
> - **XSum:** Nearest **22.2 ± 0.18**, SMILE **22.3 ± 0.13**
>
> These additional runs sharpen the interpretation. On FinPhrase, the strongest baseline remains competitive, making this a meaningful exception rather than the dominant pattern. On XSum, the performance difference indicates near-parity between methods. We will clarify this in the revision.
>
> More broadly, our claim is **not** that SMILE is strictly best on every task-LLM pair. Rather, the claim is that **joint optimization is effective overall** across datasets, LLMs, and multiple baseline families. We will make this clearer in the revision. At the same time, the overall pattern remains favorable, with stronger gains on several tasks, as well as additional support from the component ablations and black-box transfer experiments.
>
> >**W4.** White-box dependency.
> >
> **Response**: We appreciate your point and agree that our black-box results should be interpreted as **transfer to black-box LLMs**, not as direct end-to-end training on a fully opaque API model. Our goal is therefore to demonstrate **cross-LLM transferability** of the learned surrogate.
>
> We will revise the wording to make it clearer. We also believe the transfer setting remains **practically meaningful**: we show that surrogates calibrated on open-source LLMs remain effective when used to optimize prompts for black-box LLMs, where direct access to internal signals is unavailable, and API evaluation is expensive.
>
> More broadly, our main contribution is the **joint optimization framework** rather than dependence on any single basis signal. The current SMILE instantiation uses a particular set of experts because they work well empirically, but the framework does not conceptually require the same channels in every deployment. In settings without access to white-box information, one can instantiate the surrogate using alternative observable signals while retaining the same joint optimization formulation.
>
>
> >**Minors.**
> >
> **Response**: For (1) and (3), we will revise the paper accordingly. For (2), we kindly refer to the response to `Reviewer XW88` **W4**.
>
> >**Q3.** sum v.s. min.
> >
> **Response**: Thank you. We provide this ablation in our response to `Reviewer GD26` **W3**: using *sum* aggregator over the same three experts degrades overall performance. This indicates that SMILE’s gains are not only from having three scores, but also from the **EDSF structure** induced by *min*, which prevents one strong expert from masking weakness in another.

---

> > ### Author Rebuttal · Reviewer_A8CB · 2026-04-03
> >
> > The main technical concern that whether the zero-shot IG channel captures meaningful ICL interaction is convincingly addressed. The Spearman correlations between $\mathrm{IG}(e\_i, T)$and empirical single-demo ICL utility (i.e., 0.73 on XSum and 0.71 on Date) provide an evidence that IG could be a reasonable proxy for instruction-conditioned example utility, although it is measured in a zero-shot setting (see W2, Q1). The within $f\_\mathrm{SI}$ ablation (IG-only, Proto-only, and both) confirms both channels contribute, and combining them is consistently better.
> >
> > The expressiveness argument (W1) is sufficiently reframed; the theorem motivates the EDSF family, while the specific design is validated empirically. The white-box dependency (W4) is honestly acknowledged as a transfer setting. The sum-vs-min comparison (Q3) confirms the EDSF bottleneck structure contributes beyond having three complementary scores. The CIs for the two weakest Llama results are honest and clarify that SMILE's claim is about consistent overall gains, not strict dominance on every pair, which is a reasonable and accurate framing.
> >
> > I consider my concerns fully addressed and the paper a solid contribution, but the gains over baselines are incremental, the EDSF framework is borrowed, and the learned parameters are limited to scalar weights, so I maintain my _positive_ Weak Accept.

---

> > > ### Author Response · Authors · 2026-04-03
> > >
> > > Thank you very much for the careful reading and thoughtful follow-up. We sincerely appreciate your constructive suggestions throughout the review process. We are especially grateful that you found our additional analyses and clarifications helpful in addressing your concerns. Thank you again for your time and positive assessment of our work.

---

### Official Review · Reviewer_GD26 · 2026-03-12

**Soundness:** 3
**Presentation:** 4
**Significance:** 3
**Originality:** 2
**Overall Recommendation:** 4
**Confidence:** 3

**Summary:**

This paper explore the interaction between instruction optimization (IO) and in-context learning (ICL) demonstration selection, which are typically optimized separately and then combined. The authors argue that this decoupling of instructions and demonstrations does not match the their interaction in practice. To address this issue, the paper proposes SMILE, a method that jointly selects instructions and demonstrations. The authors learn an instruction-conditioned surrogate model aligned with LLM feedback and formulate it as an Extended Deep Submodular Function. This formulation captures three aspects: sample–sample coverage, sample–query relevance, and sample–instruction compatibility. It performs greedy, query-adaptive selection of instruction–demonstration pairs. Experiments across multiple datasets and LLM backbones show that SMILE consistently outperforms IO-only, ICL-only, and prior joint optimization baselines, supporting the view that effective prompting requires jointly optimizing interacting context components rather than tuning them independently.

**Compliance With Llm Reviewing Policy:**

Affirmed.

**Final Justification:**

the rebuttal addressed my main concerns

**Key Questions For Authors:**

Please refer to weakness.

**Limitations:**

Yes

**Strengths And Weaknesses:**

strengths：

Soundness

1. The method decomposes interactions into sample–sample, sample–query, and sample–instruction components using concave-over-modular structures. Greedy optimization is theoretically justified, and pairwise ranking with LLM-derived rewards is used to train the surrogate.

2. Experiments are reasonably comprehensive, covering multiple tasks and models with ablation studies validating each component.

Presentation

1. The paper is generally clear and well-structured.

Significance

1. The work addresses the problem of jointly selecting instructions and demonstrations for prompt construction. The submodular formulation provides a potentially scalable way to select prompt components without expensive LLM search.

Originality

1. The novelty mainly lies in integrating several ideas into an instruction-aware prompt selection framework. SMILE models three interactions simultaneously: sample–sample coverage, sample–query relevance, and sample–instruction compatibility.


weaknesses：

Soundness

1. In Table 1, some ICL-only methods outperform joint IO+ICL approaches. For example, on GSM8K certain ICL baselines match or exceed some joint methods, raising the question of whether joint optimization consistently provides benefits across tasks.

2. Experiments are mainly conducted on relatively small LLMs (4B, 8B, gemini flash). It remains unclear whether the improvements would remain significant for larger models with abundant knowledges, which may be less sensitive to prompt engineering.

3. The surrogate aggregates scores as f(S) = min(fSS, fSQ, fSI) The rationale for using the minimum operator instead of other form (e.g., fSS + fSQ + fSI) is not clearly justified and would benefit from theoretical or empirical explanation.

Originality

1. The SMILE framework depends on the underlying LLM during training. The surrogate is trained using teacher-forced log-likelihood from the LLM, introducing dependence on the model’s knowledge and raising questions about potential bias compared with purely heuristic baselines.

2. Prior works such as MIPRO have already explored joint optimization of prompt components. It would be helpful if the paper clarified more explicitly how its contributions differ from these existing joint optimization approaches.

---

> ### Author Rebuttal · Authors · 2026-03-29
>
> >**W1.** Table 1 explanation.
> >
> **Response**: Thank you for this observation. We agree that Table 1 shows that not every IO+ICL combination uniformly outperforms the ICL-only method. We would like to note that this phenomenon is also **not unique to our paper**: prior work [1] reports the same pattern, where APE+R.S. is sometimes worse than Random Search alone.
>
> We would emphasize that this is consistent with our main claim. Our paper does **not** argue that merely combining an instruction optimizer with an ICL selector is sufficient. Rather, the motivation for SMILE is precisely that instruction choice and sample choice interact, so **simple composition may fail to realize the full benefit of joint optimization**.  At the same time, Table 1 shows that **SMILE delivers the strongest overall results**, with especially clear gains on the more reasoning-intensive algorithmic tasks. For example, on Qwen3-4B, SMILE improves over the strongest joint/ICL baselines from 66.4 to 70 on Salient.
>
> [1] Teach better or show smarter? on instructions and exemplars in automatic prompt optimization.
>
> >**W2.** Larger LLMs.
> >
> **Response**: We agree that stronger backbones are important to test. We would like to note that we already include **Gemini2.5-Flash** in the main text and **GPT-5.2** in **Appendix A**. On both proprietary models, SMILE variants outperform the strongest baselines, suggesting that the gains are not limited to smaller models.
>
> >**W3.** Rationale for using $min$.
> >
> **Response**: Thank you for raising this. We would like to note that the use of $min$ in SMILE is **not an arbitrary design choice**, but the native EDSF-style robust conjunction. Intuitively, the three experts capture complementary requirements for a good demo set. Using $min$ enforces a bottleneck criterion: a set should not be scored highly if it is strong on one axis but weak on another. In contrast, a $sum$ form allows one large term to compensate for a weak term, which is less consistent with the “all components matter” behavior we want from the surrogate.
>
> The following experiment shows that the sum variant of SMILE reduces the average performance. This supports the intended bottleneck interpretation: the min-based EDSF better preserves the requirement that selected examples should have no major weakness.
>
> | Qwen3-4B  | GSM8K | GPQA | FP | XSum | Date | Salient | Avg |
> |-|-|-|-|-|-|-|-|
> | SMILE (Agg=sum) |  90.4 | 38.4 | 93.2 | 18.4 | 50.8 |60.0 | 58.5 |
> | SMILE  |  91.2 | 38.9 | 96.0 | 18.5 | 63.2 |70.0 | **63.0** |
>
> >**W4.** LLM dependency
> >
> **Response**: Thank you for raising this point. We agree that SMILE is **not purely heuristic**, but the LLM dependence is **localized**: teacher-forced log-likelihood is used inside the **$f^{SI}$ expert**, while the other experts are based **only** on heuristic signals. The surrogate itself is trained with an LLM-dependent ranking loss model instruction–sample compatibility, which is exactly the interaction missing from purely heuristic baselines.
>
> Also, using LLM feedback during optimization is **not unique** to SMILE. Several baselines like APE and MIPRO also rely on LLM feedback. SMILE is evaluated against both heuristic and LLM feedback-based methods and achieves the strongest overall performance. Further, Sec. 3.3 provides evidence that this LLM dependence is **not overly brittle**. The learned fSQ/fSI expert weight patterns show clear alignment across LLMs, suggesting that the surrogate captures task-specific regularities that generalize across backbones. In addition, surrogates trained with open-source LLM feedback transfer effectively to proprietary black-box models (Tables 3 and 5). This indicates that the LLM-informed signal captures meaningful information rather than merely overfitting to a single model.
>
> >**W5.** Compare with MIPRO.
> >
> **Response**: Thank you for raising this. We agree that MIPRO already explores joint optimization and want to highlight the following three contributions:
> - **Structured modeling of joint prompt utility.** Instead of searching over candidate instruction/demo configurations as MIPRO, SMILE introduces an **instruction-conditioned surrogate** that explicitly models the internal structure of prompt utility. This allows SMILE to capture *why* a demonstration set is effective for a given instruction and query, rather than only black-box search over limited combinations;
> - **A different optimization mechanism.** Because SMILE models joint utility as a structured set function, it supports **query-adaptive greedy selection** of demonstrations for each candidate instruction, whereas MIPRO does not explicitly model this interaction structure;
> - **Theory for efficient selection**: Our formulation also enables a **greedy-selection guarantee** for the learned surrogate. This sets SMILE apart from prior work that is primarily motivated by search procedures, and provides additional justification for why the method should work beyond empirical performance alone.

---

> > ### Author Rebuttal · Reviewer_GD26 · 2026-04-04
> >
> > Thanks for the response, I will maintain my positive assessment

---

> > > ### Author Response · Authors · 2026-04-06
> > >
> > > Thank you for your helpful suggestions. We are grateful that our rebuttal was able to address your concerns.

---

### Official Review · Reviewer_XW88 · 2026-03-13

**Soundness:** 3
**Presentation:** 2
**Significance:** 3
**Originality:** 3
**Overall Recommendation:** 5
**Confidence:** 4

**Summary:**

This paper focuses on the interaction between instruction optimization (IO) and in-context learning (ICL) demonstration selection, which are typically optimized independently. The authors argue that this decoupled paradigm is suboptimal because the effectiveness of demonstrations depends strongly on the instruction used. Thus, the paper proposes SMILE, a framework that jointly selects instructions and demonstrations by learning an instruction-conditioned surrogate objective based on an Extended Deep Submodular Function (EDSF). The key insight is that empirical ICL performance exhibits diminishing returns as more demonstrations are added, a property naturally captured by submodular functions. SMILE constructs a surrogate scoring function composed of three experts modeling complementary interactions: sample–sample coverage, sample–query relevance, and sample–instruction compatibility. The surrogate is trained using LLM feedback with pairwise ranking supervision and optimized using greedy selection to produce query-adaptive instruction–demonstration pairs. Experiments on six datasets and two LLM backbones show consistent improvements over IO-only, ICL-only, and joint baselines such as MIPRO. Additional experiments demonstrate transferability to black-box models such as Gemini and GPT variants.

**Compliance With Llm Reviewing Policy:**

Affirmed.

**Final Justification:**

I believe this paper effectively addresses key issues within the field of ICL and makes a substantial contribution to its advancement. In their rebuttal, the authors successfully resolved several technical concerns and clarified various ambiguities.

**Key Questions For Authors:**

* How strong is the correlation between the learned surrogate score $f(S;q,T)$ and the true downstream performance metric? Providing correlation statistics would clarify how reliable the surrogate is.
* How sensitive is SMILE to the size and diversity of the candidate instruction set? Would performance degrade if the instruction pool contained many near-duplicate instructions?
* The method assumes a relatively simple prompt structure (instruction + demonstrations). Could the framework extend to more complex prompts such as tool-use pipelines or multi-step reasoning templates?
* The surrogate is trained using feedback from one LLM and transferred to others. How stable is this transfer when the backbone models change substantially or when distribution shift occurs?

**Limitations:**

yes

**Strengths And Weaknesses:**

**Strengths**:
* The paper identifies an important limitation in existing prompt optimization pipelines. Most prior methods optimize instructions and demonstrations independently, implicitly assuming composability. The argument that instruction–demonstration interactions matter is convincing and supported with empirical observations.
* The method is tested across six datasets covering diverse tasks (math, QA, sentiment, summarization, reasoning). Results show consistent improvements over strong baselines, including APE, OPRO, and MIPRO. Cross-LLM transfer experiments to proprietary models strengthen the empirical claims.
* The paper devotes an entire subsection to efficiency analysis, which is understandable given that methods relying heavily on greedy search and submodular optimization often raise efficiency concerns.

**Weaknesses**:
* The pipeline appears somewhat complex, as each substep involves specific computations. This also makes the presentation in the Method section rather lengthy and harder to follow.
* Some relevant prior work appears to be missing. [1] studies instruction as an explicit component of ICL prompts. [2] and [3] both train new models to better capture interactions among different components of an ICL prompt, which is closely related to the motivation of this paper. In particular, [3] also considers the influence of instruction during demonstration selection. While it is not necessary to include these methods as baselines, they should be discussed in certain sections.
* Although the paper discusses efficiency, experiments use relatively small demonstration pools (e.g., 400 examples). It remains unclear how the method scales when the candidate pool becomes very large.
* The method assumes a predefined instruction set generated by another algorithm (e.g., MIPRO). The performance therefore depends heavily on the quality and diversity of this candidate pool.

[1] Li, Li, et al. "How to configure good in-context sequence for visual question answering." Proceedings of the IEEE/CVF Conference on Computer Vision and Pattern Recognition. 2024.

[2] Yang, Xu, et al. "Lever LM: configuring in-context sequence to lever large vision language models." Advances in Neural Information Processing Systems 37 (2024): 100341-100368.

[3] Li, Yanshu, et al. "Taco: Enhancing multimodal in-context learning via task mapping-guided sequence configuration." Proceedings of the 2025 Conference on Empirical Methods in Natural Language Processing. 2025.

---

> ### Author Rebuttal · Authors · 2026-03-29
>
> >**W1.** Dense Method section.
> >
> **Response**: Thank you for the helpful feedback. While SMILE has multiple subcomponents, they are introduced to capture distinct and interpretable interactions required for joint optimization, **not to add unnecessary machinery**. In practice, **the optimization is lightweight**: training only fits surrogate weights with a ranking loss, and inference uses efficient greedy selection on the surrogate with substantial offline precomputation. We will revise the presentation to make the pipeline easier to follow.
>
>
> >**W2.** Missing prior work.
> >
> **Response**: Thank you for highlighting these works. **We will add them in the revised related-work section**. Our work differs in both setting and technical contribution: we focus on **text LLMs** across diverse NLP tasks, and more importantly, our central object is **joint optimization** for a query, rather than improving sequence configuration under a fixed prompting setup. We will revise the paper to clarify this positioning.
>
>
> >**W3.** Larger pool.
> >
> **Response**: Thank you for the helpful suggestion. We agree that evaluating very large candidate pools is valuable future work. However, we note that the 400-example pool in our experiments is already comparable to the scale used in prior prompt optimization baselines: in the origianl papers, APE scores candidate instructions using 50 samples, OPRO uses 3.5% of GSM8K training data (= 260) and 20% of BBH(<100), and MIPRO uses 500 training examples.  Since most of SMILE’s surrogate terms can be precomputed offline and online search is greedy over the learned surrogate rather than repeated LLM evaluation, the method is **architecturally compatible** with larger pools.
>
>
> >**W4.** Depend on candidate pool.
> >
> **Response**: Thank you for the observation. We agree that, like other candidate-based methods, SMILE operates over a predefined instruction pool. However, SMILE is *not* tied to a specific instruction generator such as MIPRO. Our method only assumes access to a candidate instruction set; the core contribution is the joint, query-adaptive selection of instructions and demos. We additionally test candidate pools produced by APE and OPRO on Qwen3-4B, and SMILE remains **consistently strong** in performance. This suggests that the benefit comes from SMILE’s joint optimization mechanism rather than dependence on a particular upstream instruction generator.
>
> | Qwen3  | FP | Date | Sal |
> |-|-|-|-|
> | w. APE |  96.4 | 62.8 |68.8 |
> | w. OPRO  | 94.8 | 64.0 |70.4 |
> | w. MIPRO  |  96.0 | 63.2 |70.0 |
>
>
> >**Q1.** Surrogate reliability.
> >
> **Response**: Thank you for the suggestion. We agree that surrogate reliability should be assessed explicitly. Our surrogate is trained primarily for **pairwise ranking/selection**, rather than absolute score prediction: the goal is to decide which instruction–demonstration candidate is better, not to regress the final task metric itself. Thus, we measure **pairwise agreement** and find encouraging accuracy: **78.4 on GSM8K, 68.8 on GPQA, and 74.6 on Date**. These results indicate that the surrogate captures the relative ordering of candidate prompt configurations reasonably well. Since the downstream optimization only requires a reliable **ordering bias** rather than exact reward prediction, moderate but consistent pairwise agreement can already translate into better final prompt selection.
>
>
> >**Q2.** Sensitive to candidate pool
> >
> **Response**: We kindly refer to our response to `Reviewer VyvB` **W2**.
>
>
> >**Q3.** Extension discussion.
> >
> **Response**: Thank you for the question. We agree that this work focuses on a relatively simple but widely used prompt format (instruction + demos). However, the framework itself is not tied to this exact template. The main idea is to treat prompt construction as a **structured composition problem** and to learn a surrogate over the interactions among modular prompt components. In principle, this could be extended to more complex prompt programs (e.g., tool-use stages or reasoning templates) by defining additional modules and compatibility terms.  We view such extensions as promising future work rather than a claim we make here. Our main contribution is the **structured joint optimization formulation** and its effectiveness in the widely used instruction-plus-demonstration setting.
>
>
> >**Q4.** Stable transfer.
> >
> **Response**: We agree this is an important robustness question. Our current evidence is that **transfer is stable across the backbones we evaluated**: Figure 3 shows cross-LLM alignment of learned expert weights for the same task, and open-source-trained surrogates remain effective when transferred to proprietary black-box models such as Gemini2.5-Flash, with similar results on GPT-5.2 in the appendix. Our results support practical cross-backbone transfer within the evaluated settings, but broader robustness to severe backbone mismatch or shifted input distributions remains an important direction for future study.

---

> > ### Author Rebuttal · Reviewer_XW88 · 2026-04-01
> >
> > The authors' rebuttal has resolved my concerns. Thus, I have raised my score accordingly.

---

> > > ### Author Response · Authors · 2026-04-02
> > >
> > > Thank you very much for your careful reading and thoughtful feedback. We truly appreciate your time and consideration, and we are grateful that our rebuttal was able to address your concerns.

---

### Decision · Program_Chairs · 2026-04-30

**Decision:**

Accept (regular)

**Comment:**

The paper addresses the joint optimization of two prompt components (instruction + selected demos) using a learnable deep submodular function (DSF) that captures the diminishing returns of in-context learning gains as the number of demos increases. The proposed SMILE constructs a surrogate scoring function (an extended DSF) composed of three experts modeling complementary interactions: sample–sample coverage, sample–query relevance, and sample–instruction compatibility. The surrogate is trained using LLM feedback with pairwise ranking supervision and optimized using greedy selection to produce query-adaptive instruction–demonstration pairs. Experiments on six datasets and two LLM backbones show consistent improvements over IO-only, ICL-only, and joint baselines such as MIPRO. Additional experiments demonstrate transferability to black-box models, including Gemini and GPT variants.

While most reviewers agree that the joint optimization is important, and the EDSF capturing three types of interactions is novel, it was not made fully clear that the work is built on several existing works, including DSF, diminishing return properties of ICL, joint optimization from MIPRO and Mixture-of-Experts Prompts, etc. The gain of SMILE over some baselines, such as MIPRO, is incremental. Other limitations include (1) the requirement of a set of instructions; (2) unclear scalability to larger candidate sets (>400); (3) lack of justification of the minimum formulation of EDSF; (4) gap between theoretical expressiveness and empirical observations of EDSF, etc. That being said, most reviewers are satisfied with the authors‘ responses, and all reviewers vote for acceptance. After carefully reading the main techniques of the paper, despite the room for improvement, I believe this is a solid contribution to applying DSF to LLM prompt optimization, which is an underexplored topic that may have a great impact. Therefore, I recommend it for acceptance and meanwhile encourage the authors to keep improving the draft based on the rebuttal and discussion.